# Autonomous DNA nanostructures instructed by hierarchically concatenated chemical reaction networks

Jie Deng [1,3✉] & Andreas Walther [1,2✉]

Concatenation and communication between chemically distinct chemical reaction networks (CRNs) is an essential principle in biology for controlling dynamics of hierarchical structures. Here, to provide a model system for such biological systems, we demonstrate autonomous lifecycles of DNA nanotubes (DNTs) by two concatenated CRNs using different thermo-dynamic principles: (1) ATP-powered ligation/restriction of DNA components and (2) input strand-mediated DNA strand displacement (DSD) using energy gains provided in DNA toeholds. This allows to achieve hierarchical non-equilibrium systems by concurrent ATP-powered ligation-induced DSD for activating DNT self-assembly and restriction-induced backward DSD reactions for triggering DNT degradation. We introduce indirect and direct activation of DNT self-assemblies, and orthogonal molecular recognition allows ATP-fueled self-sorting of transient multicomponent DNTs. Coupling ATP dissipation to DNA nanostructures via programmable DSD is a generic concept which should be widely applicable to organize other DNA nanostructures, and enable the design of automatons and life-like systems of higher structural complexity.

[1] A3BMS Lab, Department of Chemistry, University of Mainz, Mainz, Germany. [2] Cluster of Excellence livMatS @ FIT – Freiburg Center for Interactive Materials and Bioinspired Technologies, University of Freiburg, Freiburg, Germany. [3] Present address: Department of Cancer Biology, Dana-Farber Cancer Institute, Harvard Medical School, Boston, MA, USA. ✉email: jie_deng@dfci.harvard.edu; andreas.walther@uni-mainz.de

L iving cells rely on hierarchically concatenated chemical reaction networks (CRNs) to execute biofunctions like adaptation, transport and motility as orchestrated by the dynamic assembly of the cytoskeleton for which corresponding self-assembling units are temporally activated by biological signals and fuels[1–4]. Such fuel-driven non-equilibrium systems and the organization using signaling networks in nature are role models for designing man-made energy-dissipative and self-regulating self-assembly systems and materials[5]. Importantly, the communication between CRNs of different chemistry and their correlation with coupled self-assembly processes allows to execute more complex signal transduction processes, and to regulate system transformations[6].

Taking the inspiration from nature, efforts have been made to develop synthetic fuel-driven non-equilibrium systems ranging from molecular to macroscopic levels[5,7–9]. Over the last few years, chemical fuels and CRNs have been implemented to fuel supramolecular self-assemblies for various transient systems such as for conformational switches[10], supramolecular fibrils[11–13], colloid self-assembly[14–16], hydrogel materials[13,17–21], and photonic devices[22]. Despite progress, the development of artificial non-equilibrium systems showing high structural precision and at the same time incorporating system complexities, such as multi-component assembly strategies or connection of more than one CRN, remains a challenge.

DNA has emerged as an ideal building block for self-assemblies with high structural complexity owing to its prominent molecular recognition and high programmability[23]. The emerging concept of non-equilibrium self-assembly has expanded DNA nanotechnology from molecular motors[24], walkers[25], and transport systems[24,26] to fueled systems with transient autonomous life-cycles and adaptive dynamic steady states[27–31]. For instance, by connecting DNA self-assembly systems with time-dependent degradation of RNA or DNA fuel strands through enzymatic or non-enzymatic processes, transient systems for allosteric switches[27,32], colloid clustering[29,33], and DNA nanotube self-assemblies[34] have been reported. To achieve higher system complexity, enzyme-assisted transcription/degradation reaction networks have been implemented to control dynamic DNA nanostructures[35,36]. We recently demonstrated ATP-fueled dynamization of a covalent DNA bond via an enzymatic reaction network (ERN) of concurrent ligation and cleavage, and reported details on engineering adaptiveness of the dynamics[28,37], pathway complexity[38], and light-modulated behavior on a polymer level[39]. Subsequently, we reported ATP-fueled transient multivalency to regulate autonomous colloidal self-assembly and all-DNA coacervates on a multicomponent systems level by programming orthogonal molecular recognitions therein[40,41]. We could establish a communication between ATP-fueled ERNs with toehold-mediated DNA strand displacement (DSD) to realize ATP-fueled DSD cascades and automatons[42]. However, autonomous dynamic control of higher-order DNA nanostructures by multiple concatenated distinct CRNs is still rare. A relevant report is the autonomous control of DNA nanotube assembly via RNA transcription using dNTPs by Franco and coworkers, wherein RNA is produced to disassemble DNA nanotubes (DNTs) and subsequent RNA degradation allows for re-assembly of DNTs[36]. The signal RNA (B) is generated in a temporally controlled fashion and can be coupled to an upstream oscillator, but its presence follows an A (dNTPs; fuel) to B (RNA) to C (waste) pathway. Hence, the signal generation is rather energetically downhill, and after its conversion to waste it cannot be reused. Here, we intend to develop ATP-fueled hierarchical self-assembly in an energetically up-hill fashion using cyclic reaction network, in which the ATP-fueled CRN module generates a signal that is transduced via a different CRN to achieve a hierarchical structure downstream. The signal is recovered after fuel consumption and can be reused (A to B to A) by a fuel (ATP) that is not connected to the structure formation directly. Success in this direction would help to better mimic the hierarchical non-equilibrium systems in nature and provide more insights into future life-like materials design.

In the present study, we demonstrate ATP-fueled autonomous DNT assembly regulated by DSDs that are induced and regulated upstream by an ERN of concurrent ATP-powered ligation and restriction of DNA components. The ATP fuel is consumed in the ERN, while the transient output regulates downstream DSD and DNT assembly in a hierarchical network fashion. We demonstrate two approaches for the DNT assembly: (1) indirect control of DNT assembly by ATP-fueled DSD cascades and (2) direct control of DNT assembly by ATP-fueled transient ligation and ensuing DSD on the assembling tiles. The direct control via ATP-fueled transient ligation on the assembling tiles requires significantly less DNA species than the ATP-fueled DSD cascade strategy because it allows to reduce layers of DSD reactions. By programming orthogonal molecular recognition into parallel systems, we further achieve ATP-fueled and transiently self-sorting DNTs. We envision that our strategies to control ATP-fueled, non-equilibrium DNA nanostructure self-assembly will inspire to develop life-like materials with easily reconfigurable responses as well as hierarchical control mechanism on a structural and network level.

## Results

**General concept for ATP-driven autonomous DNA nanotube self-assemblies.** Building on our previous design[42], we first concatenate an ATP-fueled ERN of an ATP-powered ligation/restriction system of DNA components (upstream) with transient DSD cascades using energy gains provided in DNA toeholds (intermediate layer) to regulate DNA nanostructure formation (downstream) as shown for DNTs (Fig. 1a). Both the ATP-driven network and the DSD operate on a molecular level, while the DNT formation then hierarchically reports on a nanostructural level as seen by microscopy. In strategy 1, the ATP-powered ligation via T4 DNA ligase induces the release of an output strand by transiently biasing the energy landscape of a strand displacement reaction due to covalent linkage of an input strand. Downstream Output Regulator removes Inhibitor 1 from Inactive tile 1 and triggers DNT self-assembly (Fig. 1a). Consequently, the DNT self-assembly is ultimately coupled upstream to the ATP-driven dynamic steady state (DySS) with a transient lifecycle, giving rise to a synthetic hierarchical non-equilibrium DNT self-assembly system powered by ATP. In strategy 2, the release of the output strand liberates a free single-stranded (ss) DNA domain on the previously inactivated tileso that assembly takes place, while the ligated part sticks out of the assembling tile. The output strand is of no immediate relevance to the DNT formation (Fig. 1b). The concurrent endonuclease restriction induces the generation of an intermediate that can be dissociated by the free inhibitors in the system. In the activated tile, a 5 nt ssDNA spacer encoded in black between the dangling double-stranded (ds) DNA and the ssDNA assembling domain is designed to facilitate the sticky end interactions for self-assembly. The dynamic DSD is maintained in an ATP-driven DySS through the ERN of concurrent ATP-fueled ligation and restriction[28], and the concatenated CRNs are bound in their behavior to the consumption of ATP, leading to a programmable DSD and DNT lifetime. By programming orthogonal molecular recognition in multiple self-assembly systems, we further achieve ATP-fueled transient self-sorting DNTs on a multicomponent systems level.

## Strategy 1: Indirect Activation for ATP-Fueled Transient DNA Nanotube

**a**

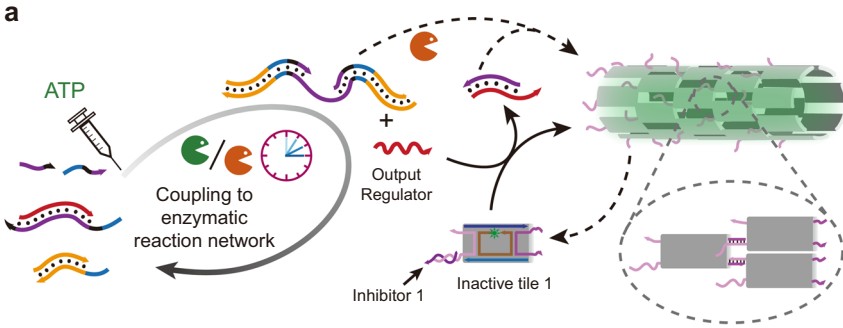

## Strategy 2: Direct Activation for ATP-Fueled Transient DNA Nanotube

**b**

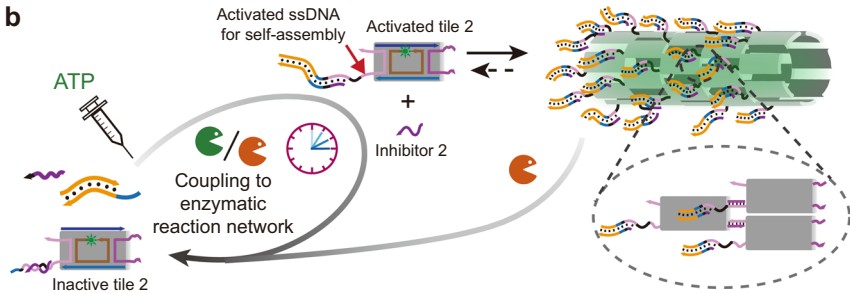

**Fig. 1 Proposed control mechanisms for autonomous control of DNT self-assemblies. a** ATP-fueled transient DNT self-assembly regulated by ATP-fueled DSD cascades. The 3′ ends are marked with an arrow. The design of the double-crossover tile and DNA nanotube is adapted from a previous report with permission[36]. **b** Schematic representation of autonomous control of DNT self-assembly by direct ATP-fueled DSD on the assembling tile to activate the inactive tile.

**ATP-fueled transient DNA strand displacement**. We build on our previous ATP-driven DSD circuits studied exclusively on a molecular level[42], but made critical design improvements by designing a Substrate 1 that can be invaded from both sides via ATP-powered ligation (using Complex 1, Input 1 and Input 2) to allow for a longer Output Regulator strand to be ejected from Substrate 1. This dual-side invasion to release longer output strands provides a higher thermodynamic push and higher programmability of the downstream DSD cascades. The ligation transiently biases the energy landscape of the strand displacement circuit. Note that Input 1 and Input 2 must be relatively short sequences (with respect to the operating temperature) so that the DSD only happens when ATP-powered ligation proceeds.

We first introduce the ATP-fueled dual-sided invasion for transient DSD (Fig. 2a). The system contains two dsDNA (Complex 1 and Substrate 1), and two ssDNA (Input 1 and Input 2). Substrate 1 carries 3-nucleotide (nt) and 2-nt toeholds (coded in black color) on its left and right sides, respectively, and it has a duplex part of 18 base pairs (bp) inbetween (details of the sequences in Supplementary Table 1). Before adding ATP, Input 1 and Input 2 cannot kick out any ssDNA output from Substrate 1 due to unstable hybridization of the longer DNA strand in Substrate 1 with Input 1 (7 bp), and with Input 2 (8 bp) (melting temperature ($T_{\mathrm{m}}$) < 25 °C, determined by NUPACK). However, after adding ATP, two Complex 1 molecules are covalently ligated with Input 1 and Input 2 to Substrate 1 to generate Intermediate 1, which activates chain migration and leads to release of Output 1 and formation of Complex 2. Simultaneously, BsaI cleaves the newly formed Complex 2 to regenerate Complex 1 and Intermediate 2 which dissociates to three ssDNA strands due to its low $T_{\mathrm{m}}$, and thereby reproduces Input 1 and Input 2, as well as Substrate 1 by re-hybridizing with Output 1. The connected CRNs and thus the DSDs are maintained in an ATP-driven DySS

with an ATP-dependent lifetime. Output 1 will later be the key component to couple to the DNT assembly.

A basic experimental protocol for the system design is as follows: The experiments were carried out at 25 °C using 20 μM Complex 1, 5 μM Substrate 1, 10 μM Input 1, 10 μM Input 2, 0.8 Weiss units (WU) μL$^{-1}$ T4 DNA ligase, 1.5 units (U) μL$^{-1}$ BsaI, and 40 μM ATP. Agarose gel electrophoresis (AGE) analyses of time-dependent aliquots of the systems reveals the transient nature of the system regarding formation of transient intermediates and their yields (Fig. 2b). The dynamic system shows a fraction of DSD of ca. 50 % at its DySS, as calculated by the yield of Complex 2 compared to Substrate 1 (Supplementary Fig. 1a). It is worth noting that the fraction of DSD can be further tuned by the ratio of both enzymes[42]. Since AGE is not an ideal method for determining the lifetimes due to limited temporal resolution, we further monitored the transient DSD by an allosteric switch of the output strand (Fig. 2c). In this design, Substrate 1 is replaced by Substrate 2 which carries a fluorophore and a quencher at the 3′ and 5′ ends of the output strand, respectively, allowing for Förster resonance energy transfer (FRET) when in close proximity. No DSD happens in absence of ATP. The system shows a high fluorescence intensity (FI) because the fluorophore and quencher are far away from each other. After adding ATP, Output 2 carrying the FRET pair is expelled from Substrate 2 and forms a coiled structure due to its less rigid nature compared to its dsDNA state[27], which leads to a close proximity of the fluorophore and quencher, and, thus, to a decreased FI. Simultaneously, the BsaI restriction leads to regeneration of Substrate 2, and, thus, recovery of the FI. The transient allosteric switch is maintained in an ATP-driven DySS in the ERN. After ATP is consumed, the system returns to its original state. The time-dependent FI traces reveal accurate lifetimes of the ATP-fueled allosteric switch and the DSD (Fig. 2d). By increasing the ATP concentration from 40 to 80 μM, the lifetime for the

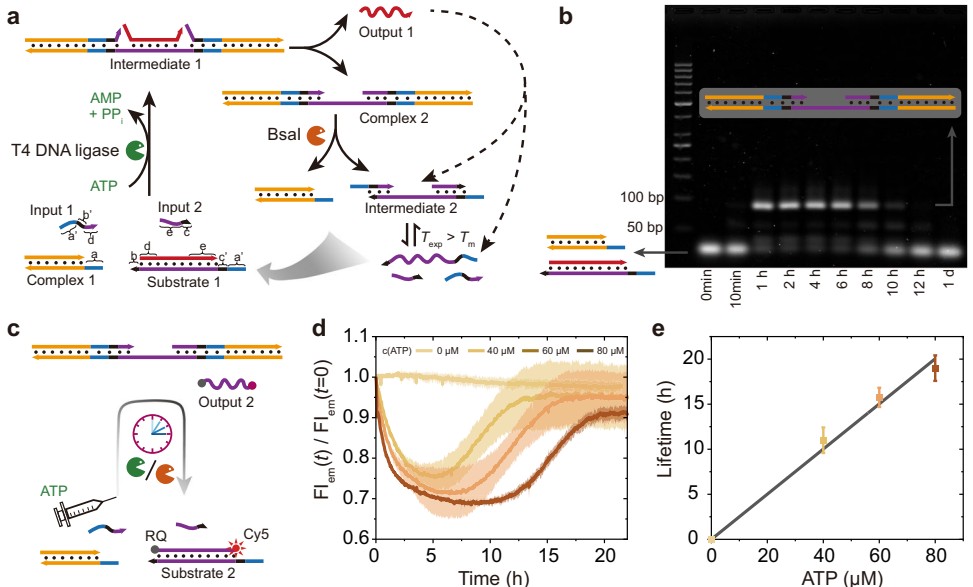

**Fig. 2 ATP-fueled dual invasion transient DSD with transient release of a long output strand. a** Schematic representation of ATP-fueled transient DSD. Experimental temperature ($T_{exp}$) = 25.0 °C, $T_m$ of Intermediate 2 = 20.0 °C (by NUPACK simulation). **b** AGE analysis of ATP-fueled DSD (3 wt.%, 80 V, 2.5 h). **c** Schematic representation of transient DSD with a FRET pair for in-situ DSD readout via an allosteric switch. **d, e** Programmable lifetimes seen by time-dependent FIs at different ATP concentrations. Shaded areas in **d** and error bars in **e** correspond to the standard deviation of duplicate measurements. Conditions: **a** 25 °C, 20 μM Complex 1, 5 μM Substrate 1, 10 μM Input 1, 10 μM Input 2, 0.8 WU μL$^{-1}$ T4 DNA ligase, 1.5 U μL$^{-1}$ BsaI, and 40 μM ATP. **c** The same as **a** except that Substrate 1 is replaced by Substrate 2, and ATP concentration varies from 40 to 80 μM.

transient DSD increases from ca. 12 to 19 h, as calculated from the point showing 80 % recovery of the decreased FI (Fig. 2d, e; Supplementary Fig. 1b). The slightly decreased FI (ca. 2.5 %) for the system without ATP is due to dye bleaching during the measurements. It is worth noting that the ATP-fueled allosteric switch only functions in presence of both Input 1 and Input 2, and missing any one of the inputs cannot deliver a successful DSD (Supplementary Fig. 1c).

**ATP-fueled transient DNA strand displacement cascades.** In this part, we further demonstrate how the output from the above ATP-fueled DSD can regulate downstream toehold-mediated DSD to realize ATP-fueled DSD cascades. Fig. 3a shows such a design, where we incorporate a dsDNA (Substrate 3) reporter for the DSD cascades using Output 1 as an input. ATP-powered ligation of Complex 1, Substrate 1, Input 1, and Input 2 activates the first layer DSD, generating Output 1 and Complex 2. Output 1 further kicks out Output 3 from Substrate 3 via toehold-mediated DSD accompanied by Complex 3 formation, which can be monitored by FI increase of Output 3 due to an incorporated FRET pair in Substrate 3. Substrate 3 is composed of a fluor-ophore strand and a quencher strand. Simultaneously, the BsaI restriction of Complex 2 leads to regeneration of Complex 1 and Intermediate 2 that dissociates to three ssDNA strands, triggering backward DSD cascades to regenerate Substrates 1 and 3.

The experiments were carried out at 25 °C using 20 μM Complex 1, 5 μM Substrate 1, 1 μM Substrate 3 (fluorophore strand to quencher strand ratio, 1:2, see Fig. 3a), 10 μM Input 1, and 5 or 10 μM Input 2 (meaning one or two equivalents in reference to Substrate 1), as well as 40 μM ATP. Time-dependent FI traces allow an in-situ monitoring of the ATP-fueled transient DSD cascades (Fig. 3b). Transient increase of FI is achieved by ATP-fueled transient DSD cascades with varied concentration of Input 2, where two equivalent amounts of Input 2 show slightly faster kinetics for the DSD cascades. Critically, due to the interference of the regeneration of Substrate 3 by the hybridiza-tion between Output 3 and the long ssDNA dissociated from

Intermediate 2, two equivalents of the quencher strand are needed for swift Substrate 3 regeneration. By using only one equivalent of the quencher strand, the FI recovery is quite time-consuming (Fig. 3c vs. Fig. 3b), because the quencher strand has only one more nucleobase to compete with the long ssDNA from Intermediate 2 to hybridize with Output 3. Additionally, two equivalents of the quencher strand in Substrate 3 can also better quench the FI, leading to a lower FI at $t = 0$ and thus higher increase of normalized FI at DySS. (Fig. 3b, c). Furthermore, repeated activation of the transient DSD cascades is possible by repeatedly adding ATP fuel. The experiments were carried out similar as above by using two equivalents of both Input 2 and quencher strand, and 20 μM ATP for each transient lifecycle. Time-dependent FI measurements verify the capability to repeatedly fuel the system (Fig. 3d), which shows prominent stability for multiple reproducible transient lifecycles.

Furthermore, we investigated whether these ATP-fueled DSD cascades can also operate at 37 °C (Supplementary Fig. 2a). For this case, Substrate 1, Input 1, Input 2, and Substrate 3 were replaced by Substrate 4, Input 3, Input 4, and Substrate 5, respectively, for longer lengths due to higher allowed $T_m$ for the Intermediate from BsaI restriction. By increasing the ATP concentration from 40 to 80 μM, the lifetime for the transient DSD cascades increases from ca. 7 to 11 h, as calculated from the point showing 80% recovery of the increased FI (Fig. 3e, f). The shorter lifetime and lower FI increase compared to the ATP-fueled transient DSD cascades at 25 °C are attributed to higher BsaI activity at 37 °C[40]. To get a comparable ratio of DSD, one would need to increase T4 DNA ligation concentration or decrease the BsaI concentration. Moreover, the results of the experiments at 37 °C by using one equivalent amount of the quencher strand also shows slow recovery of FI agreeing with the results at 25 °C (Supplementary Fig. 2b).

**Autonomous DNT self-assembly by ATP-fueled transient DSD cascades (Strategy 1).** DNTs have emerged as a promising fibrillar self-assembly for constructing artificial programmable

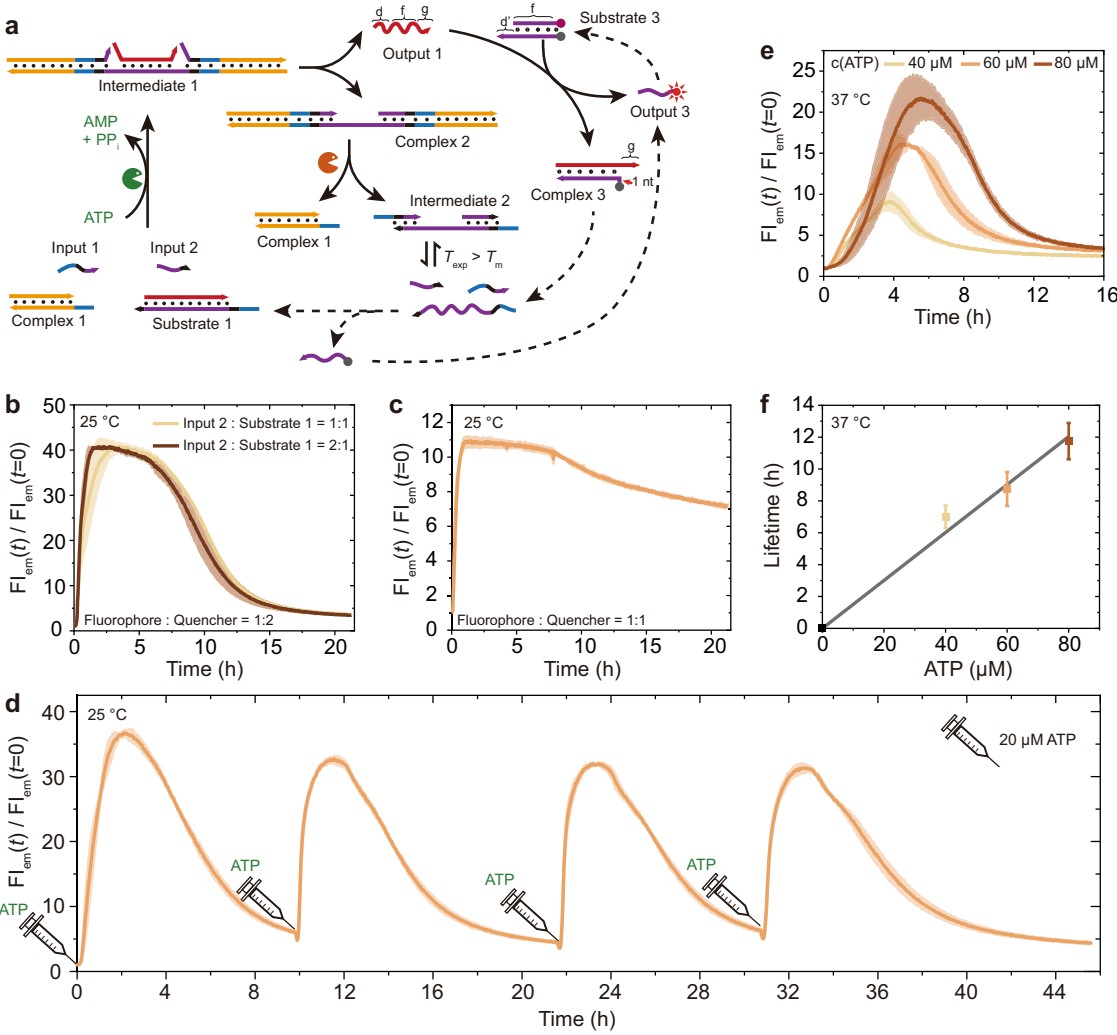

**Fig. 3 ATP-fueled transient DSD cascades using the dual invasion strategy. a** Schematic representation of ATP-fueled transient DSD cascades. **b** Time-dependent FI at different amount of Input 2. Substrate 3 has a ratio of fluorophore strand to quencher strand of 1:2. Shaded areas correspond to the standard deviation of duplicate measurements. **c** Time-dependent FI of the system via Substrate 3 with a ratio of fluorophore strand to quencher strand of 1:1. Shaded area corresponds to the standard deviation of duplicate measurements. The lower increase and the slower kinetics are due to low efficiency of Substrate 3 formation at equimolar ratio. **d** Multiple activation of the system by repeatedly adding ATP. Shaded area corresponds to the standard deviation of duplicate measurements. **e, f** Programmable lifetimes for the systems running at 37 °C seen by time-dependent FIs at different ATP concentrations. Shaded areas and error bars correspond to the standard deviation of duplicate measurements. Conditions: **b** 25 °C, 20 μM Complex 1, 5 μM Substrate 1, 1 μM Substrate 3 (fluorophore strand to quencher strand ratio, 1:2), 10 μM Input 1, 5 or 10 μM Input 2, 0.8 WU μL$^{-1}$ T4 DNA ligase, 1.5 U μL$^{-1}$ Bsal, and 40 μM ATP. **c** The same as **b** except that Substrate 3 with one equivalent amount of quencher strand and 10 μM Input 2 were used. **d** The same as **b** but using 10 μM Input 2, and the system was repeatedly fueled with 20 μM ATP for four transient lifecycles. **e** 37 °C, 20 μM Complex 1, 5 μM Substrate 4, 1 μM Substrate 5 (fluorophore strand to quencher strand ratio, 1:2), 10 μM Input 3, 10 μM Input 4, 0.8 WU μL$^{-1}$ T4 DNA ligase, 1.5 U μL$^{-1}$ Bsal, and varied concentration of ATP.

cytoskeletal networks[43]. Inspired by the hierarchical non-equilibrium systems in nature, we hypothesized that the downstream DSD could connect our ATP-fueled, concatenated CRNs to DNT self-assembly. DNTs are conceptually derived from a double-crossover DNA tile known as DAE-E, first reported by Seeman and coworkers[44,45], whereas Rothemund and coworkers firstly reported DNT assembly[46]. Towards our first strategy, we adapted the reengineered tiles from Franco and Ricci groups to our ATP-fueled system[34,36]. The assembling tile is made of five distinct ssDNA strands containing four sticky ends (each of 5 nt). These DNA tiles are able to self-assemble into micrometer-scale hollow tubular structures at room temperature via sticky end interactions. However, the addition of inhibitor strands can efficiently deactivate the tiles for self-assembly (Fig. 4a; Supplementary Fig. 3a). By coupling the inhibitor strands to a transient

DSD removal reaction, we surmised that the inactive tiles could be reversibly activated and deactivated by Activator and Deactivator strands, respectively (Fig. 4a). Indeed, confocal laser scanning microscopy (CLSM) images in Fig. 5a confirm the formation of DNTs at 25 °C after adding 3 μM Activator strands (=Output 1) to a solution containing 0.5 μM Inactive tile 1 deactivated by 1 μM Inhibitor 1. The DNTs disassemble after adding 5 μM Deactivator strands. The histogram in Fig. 5d shows the length distribution of the assembled DNTs with a mean length, $<L> \approx 3.95$ μm after reacting for 30 min.

Next, we combined the DSD-mediated DNT self-assembly with our ATP-fueled DSD cascades. Fig. 4b shows such a design. After adding ATP, the Output 1 expelled from Substrate 1 via ATP-fueled DSD acts as Activator strand for the Inactive tile 1, giving rise to generation of Complex 6 and the Activated tile 1 that can

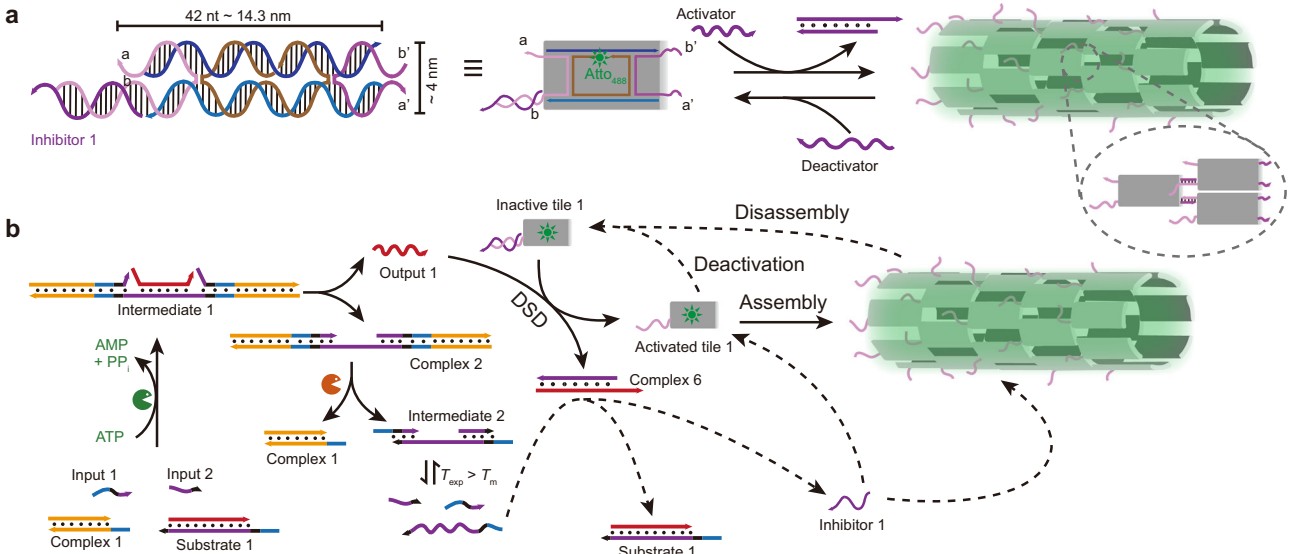

**Fig. 4 Dynamic control of DNT self-assemblies. a** Schematic representation of Inactive tile 1, and its activation for DNT self-assembly and then deactivation by Activator and Deactivator strands, respectively. **b** Schematic representation of autonomous dynamic control of DNT self-assembly by ATP-fueled transient DSD cascades.

self-assemble into DNT via sticky end interactions. Simultaneously, Intermediate 2 generated from restriction of Complex 2 dissociates to Input 1, Input 2, and another long ssDNA which triggers backward DSD with Complex 6, giving rise to regeneration of Substrate 1 and Inhibitor 1. Then, Inhibitor 1 further deactivates the assembled DNTs to reset the whole system. It is worth noting that the DNTs can be degraded both from the end or any position in the middle when invaded by the inhibitor strands.

We first performed simple enzymatic switching experiments without BsaI being present initially to show static ATP-fueled ligation-induced DNT self-assembly, while, later on, addition of BsaI leads to degradation of the assembled DNT structures (Fig. 5b; Supplementary Fig. 3b). The system contains 0.5 μM Inactive tile 1 (with 1 μM Inhibitor), 20 μM Complex 1, 5 μM Substrate 1, 10 μM Input 1, 10 μM Input 2, 0.8 WU μL$^{-1}$ T4 DNA ligase, and 60 μM ATP. The assembled DNTs show a $<L> \approx 4.17$ μm after reacting for 2 h. Lower magnification CLSM images in Supplementary Fig. 4 show lower yield of DNTs by ATP-powered DSD compared to the system directly activated by Activator strands. Afterwards, BsaI was added to the above reaction solution at a final concentration of 1.5 U μL$^{-1}$, and the DNTs disassemble after one day.

To move beyond this rather classical step-by-step switching, for the autonomous control of DNT self-assembly, BsaI was added to the system prior to the ATP addition. The concentrations of all species are identical as above except that 40 μM ATP was added as fuel for the autonomous process. CLSM measurements at different time points reveal the transient nature of the ATP-fueled transient DNT self-assembly regarding length and yield (Fig. 5c). Before adding ATP, the tiles are deactivated for self-assembly and no structure can be observed in CLSM. After adding 40 μM ATP, DSD cascades occur in the system and the inactive tiles are activated in the DSD reaction network, giving rise to DNT self-assembly. After 30 min, some DNTs with a $<L> \approx 0.9$ μm are observed (Fig. 5c, g). Subsequently, further growth occurs and a relatively steady plateau with a mean length of 3.5 μm is reached after ca. 2 h (Fig. 5f, g). Concurrently, the average length of DNT per square micrometer reaches to a plateau of 0.045 μm μm$^{-2}$ (Fig. 5g), which is used as an indicator for the yield of the DNTs. After ATP is consumed, the DNTs are

degraded. Compared to the static systems (sequential addition of Activator strands and ATP-powered ligation only) the DNTs in the ATP-fueled transient DSD cascades show shorter $<L>$ and lower yield (Supplementary Fig. 4c). This can be explained by less Activator strands (= Output 1) in the dynamic system. Additionally, repeated activation of the system can be achieved by resupply of 40 μM ATP after the first round of DNT degradation, leading to a second transient lifecycle for the DNT self-assembly (Fig. 5c, g; Supplementary Fig. 5). The lower magnification CLSM images in Supplementary Fig. 6 show larger fields of view for the transient DNT self-assemblies to underscore the homogeneity of the systems. It is worth noting that the lifetime for the DNA nanotube can in principle be programmed by the ATP concentration due to its direct connection to ATP-dissipative cyclic ERNs[40]. In this study, we simply fueled the system with sufficient ATP (2 times of needed ATP for full ligation) to ensure a DySS of the DNA nanotube assembly. The system cannot enter into its DySS when there is not enough ATP available for the ligation[40].

**Autonomous DNT self-assembly controlled by direct ATP-fueled DSD on the assembling tile (Strategy 2).** To realize the second strategy towards autonomous control of DNT formation via direct manipulation of the assembling tiles, we designed a system, in which Complex 1 and Input 5 sequentially ligate with the Inactive tile 2, giving rise to the release of Inhibitor 2 and formation of Activated tile 2 (Fig. 6a; Supplementary Fig. 7). The activated tiles then self-assemble into DNT via sticky end interactions by liberation of a ssDNA part originally shielded by Inhibitor 2. This results in a dangling dsDNA part on the DNTs that is covalently attached by ligation. However, it is worth noting that the dangling dsDNA does not affect the DNT self-assembly as there is a 5-nt ssDNA spacer between the dsDNA stem and corresponding sticky end (Fig. 1b; Supplementary Fig. 7). Simultaneously, the BsaI restriction occurs on both the activated tiles and the assembled DNTs. The cleavage of Complex 1 from the DNTs leads to dissociation of Input 5 from the DNTs due to its low $T_m$ (6 bp). This leads to the formation of DNTs with dangling ssDNA strands that can act as toeholds for re-hybridization with Inhibitor 2 to degrade the DNTs. The system can be perceived as more complicated as the enzymes now

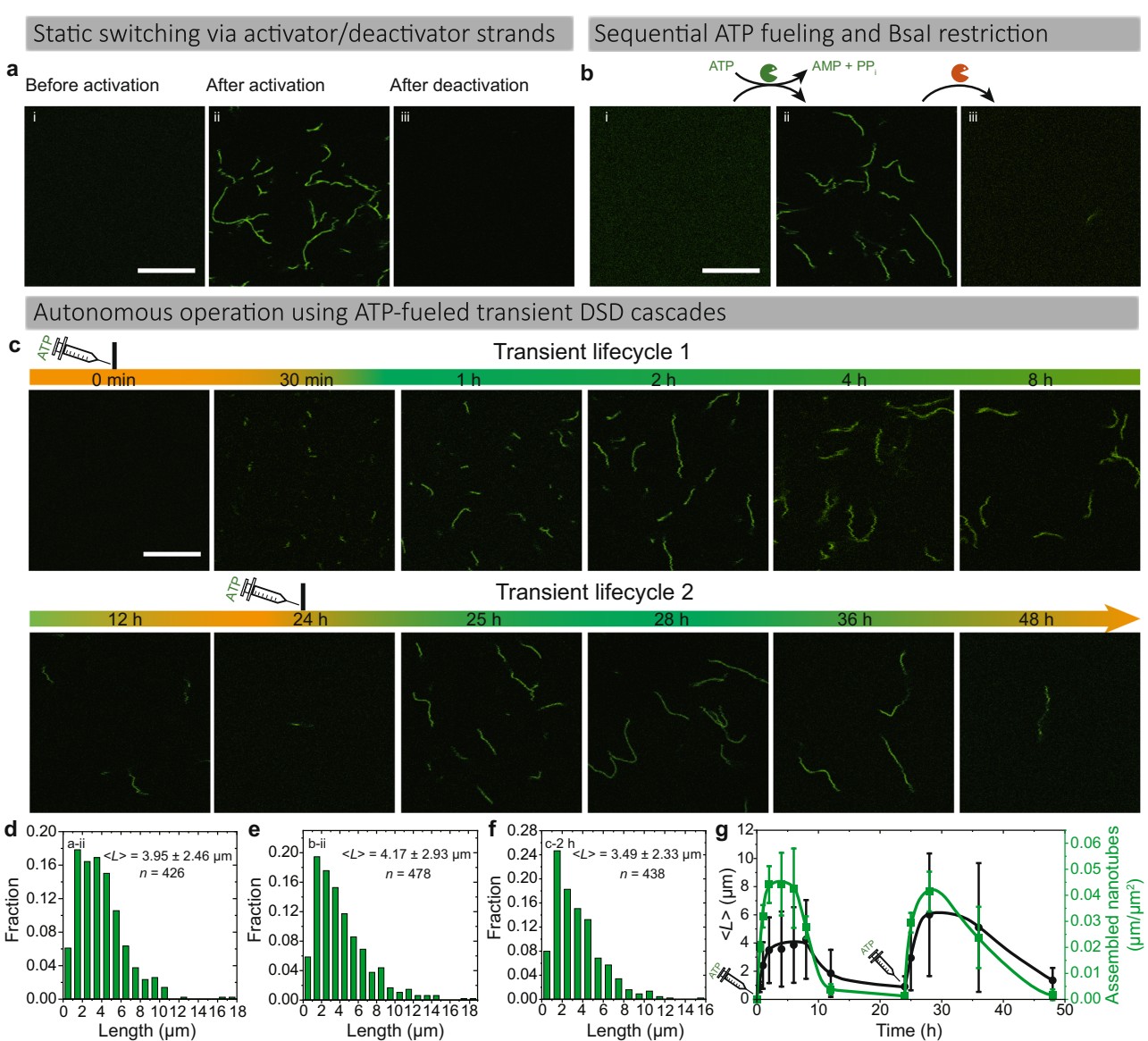

**Fig. 5 From sequential switching of DNTs to autonomous ATP-fueled systems with transient lifecycles. a** CLSM measurements of DNTs (i) before and (ii) 30 min after adding Activator strand (= Output 1), and (iii) 30 min after adding excess Deactivator strand for deactivation. Scale bar: 10 μm. **b** CLSM measurements of DNT assembly and disassembly by sequential ATP-powered ligation and BsaI-induced restriction. Scale bar: 10 μm. **c** Time-dependent CLSM measurements of ATP-fueled transient DNT self-assembly. Scale bar: 10 μm. Histograms of DNT length measured from CLSM images for (**d**) static activation by activator strands, (**e**) ATP-fueled ligation only, and (**f**) 2 h of ATP-fueled transient self-assembly. $<L>$ indicates mean DNT length. **g** Time-dependent $<L>$ and yield of the ATP-fueled transient DNT self-assembly. Error bars correspond to the standard deviation of at least three random fields of view each with an area of $185.4 \times 185.4\ \mu m^2$ from duplicate experiments. Conditions: **a** 25 °C, 0.5 μM Inactive tile 1 (with 1 μM Inhibitor 1), followed by adding 3 μM Activator strands (= Output 1), and then 5 μM Deactivator strands. **b** 25 °C, 0.5 μM Inactive tile 1 (with 1 μM Inhibitor 1), 20 μM Complex 1, 5 μM Substrate 1, 10 μM Input 1, 10 μM Input 2, 0.8 WU μL$^{-1}$ T4 DNA ligase, and 60 μM ATP, followed by adding 1.5 U μL$^{-1}$ BsaI. **c** 25 °C, 0.5 μM Inactive tile 1 (with 1 μM Inhibitor 1), 20 μM Complex 1, 5 μM Substrate 1, 10 μM Input 1, 10 μM Input 2, 0.8 WU μL$^{-1}$ T4 DNA ligase, 1.5 U μL$^{-1}$ BsaI, and 40 μM ATP. After 24 h, another 40 μM ATP was added for the second transient lifecycle.

need to operate on the tiles and the DNTs, and may not only need to be present in solution.

The experiments were carried out using 1 μM Inactive tile 2 (with 1 μM Inhibitor), 5 μM Complex 1, 5 μM Input 5, and 8 μM ATP. Time-dependent CLSM measurements visualize the ATP-fueled transient DNT self-assembly. Before ATP addition, all the assembling tiles are deactivated and no structure can be observed in CLSM. After ATP addition, the inactive tiles are activated for DNT self-assembly. After 30 min, DNTs with a $<L> \approx 1.2\ \mu m$ are observed (Fig. 6b, d). Interestingly, the DNT growth at the beginning is faster than that for the system controlled by ATP-fueled DSD cascades (with a $<L> \approx 0.9\ \mu m$ at 30 min), which is

due to efficient, direct release of Inhibitor 2 during the activation of tile 2 via one-layer direct ATP-fueled DSD. Subsequently, further growth occurs and a relatively steady plateau with a $<L> \approx 2.2\ \mu m$ is reached after ca. 1 h (Fig. 6d). Concurrently, the average length of DNTs per square micrometer reaches to 0.13 μm μm$^{-2}$ (Fig. 6d), which is thrice the yield by ATP-fueled DSD cascades above (ca. 0.04 μm μm$^{-2}$), although the concentration for the assembling tile is only doubled. Histograms show length distribution of the DNTs (Fig. 6c; Supplementary Fig. 8). Compared to the system controlled by DSD cascades (above), shorter DNTs are observed at the DySS, which is attributed to the more dynamic nature of the DNTs due to direct restriction on the

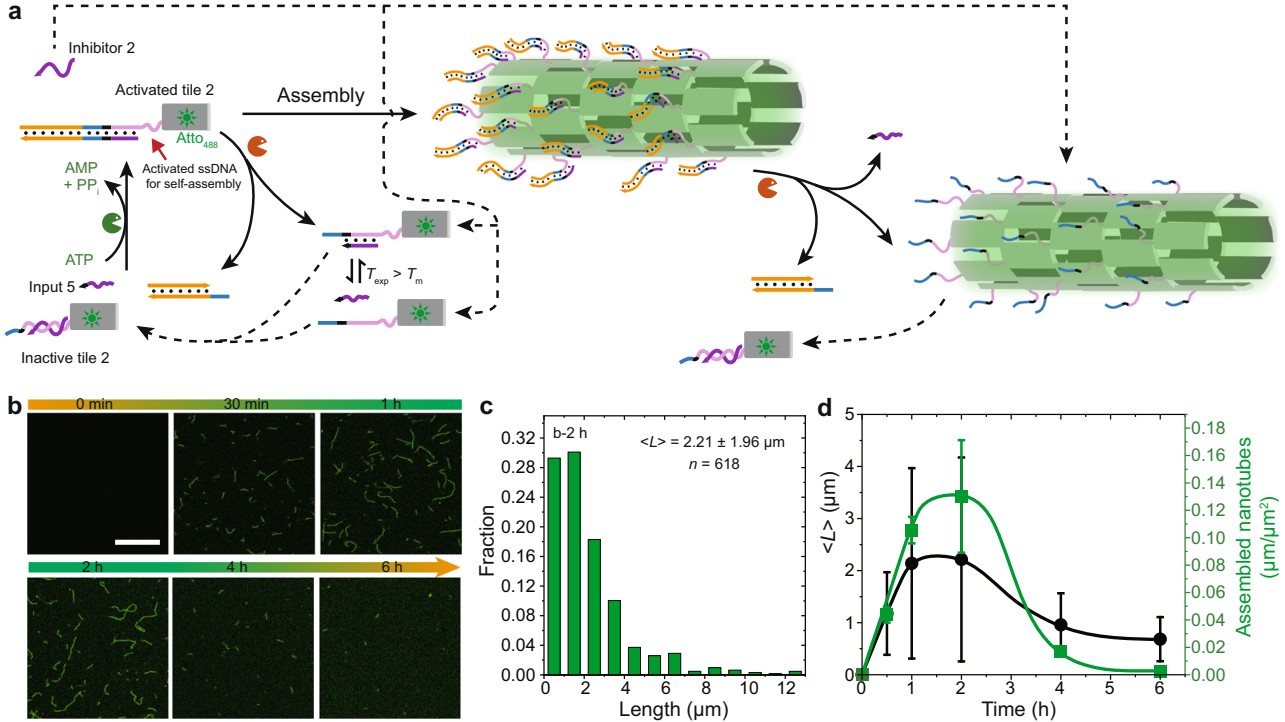

**Fig. 6 Autonomous DNTs controlled by direct ATP-fueled DSD on the assembling tile. a** Schematic representation of ATP-fueled transient DNT by direct one layer ATP-fueled transient DSD on the assembling tile. $T_{exp}$ = 25.0 °C, $T_m$ of Intermediates from restriction = 20.2 °C (seen by NUPACK simulation). **b** Time-dependent CLSM measurements of ATP-fueled transient DNTs. Scale bar: 10 μm. **c** Histogram of DNT length distribution at 2 h measured from CLSM images. < L > indicates mean DNT length. **d** Time-dependent < L > and yield of the ATP-fueled transient DNT self-assembly. Error bars correspond to the standard deviation of at least three random fields of view each with an area of 185.4 × 185.4 μm² from duplicate experiments. Conditions: 25 °C, 1 μM Inactive tile 2 (with 1 μM Inhibitor 2), 5 μM Complex 1, 5 μM Input 5, 0.8 WU μL⁻¹ T4 DNA ligase, 1.5 U μL⁻¹ BsaI, and 8 μM ATP.

DNTs and also a higher ratio of BsaI to cleavable sequences (1.5 U μL⁻¹ to 1 μM versus 1.5 U μL⁻¹ to 10 μM). The higher DNT yield (as judged by the length per μm²) is attributed to more efficient tile activation. After ATP is consumed, the DNTs are degraded.

**ATP-fueled transient self-sorting DNA nanotubes.** In this last part, we further introduce ATP-fueled, transiently self-sorting DNTs as a model system for fuel-driven multicomponent self-sorting in DNA-based fibrillar nanostructures. Critically, strategy 2 shows great advantage over strategy 1 for multicomponent system, as the short sticky ends on Substrate 1 can be easily blocked by the other ssDNA overhangs in the system via covalent ligation in case of increased number of DNA species. Following strategy 2, we achieve this by programming orthogonal molecular recognition segments at the toeholds of both inputs and at the sticky ends for DNT self-assemblies (Fig. 7a; Supplementary Fig. 9a). Complex 1 is firstly ligated with Inactive tile 2 or Inactive tile 3. Then, Input 5 and Input 6 are ligated to Inactive tile 2 and Inactive tile 3, respectively, leading to release of Inhibitor 2 and Inhibitor 3, and, thus, to the formation of Activated tile 2 and Activated tile 3. Afterwards, both activated tiles undergo narcissistic self-assembly due to orthogonal molecular recognition in the sticky end interactions, giving rise to the formation of self-sorting DNTs, DNT-A and DNT-B. Simultaneously, BsaI restriction on both types of DNTs triggers DNT degradation by the corresponding inhibitors. Tile 2 and tile 3 are fluorescently labeled with Atto₄₈₈ and Atto₆₄₇, respectively, to allow for visualization of both DNTs via CLSM.

Time-dependent CLSM measurements visualize the ATP-fueled, transiently self-sorting DNTs. No structure is observed in CLSM before ATP injection. After addition of ATP and

reaction for 30 min, some short green and red DNTs can be observed (Fig. 7b), which show similar < L > ≈ 1.4 and 0.9 μm, respectively (Fig. 7c). After 2 h, both DNTs grow to ca. 3 μm with yields of 0.07 and 0.05 μm μm⁻² for DNT-A and DNT-B, respectively (Fig. 7b, c). After ATP is consumed, both systems reset to the original disassembled state. More details about the DNT length distribution for the transiently self-sorting system are shown in Supplementary Fig. 9b.

## Discussion

In this work, we have introduced the concept of hierarchical control of non-equilibrium self-assembly of DNA nanostructure, where the DNA nanostructure self-assembly is regulated by two distinct and hierarchically concatenated CRNs, serving as a model system to mimic and understand hierarchical non-equilibrium systems in nature. The dynamic DNA nanostructure self-assembly is in principle regulated by DSD cascades, whose operation is coupled upstream to an ERN of concurrent ATP-powered ligation using T4 DNA ligase and BsaI-induced restriction of DNA components. The DySSs for both DSD cascades and DNA nanostructure self-assembly are synchronized in the ATP-fueled ERN. Hence, the ATP concentration regulates the lifetime for both downstream processes, realizing an ATP-fueled autonomous hierarchical non-equilibrium system.

We introduced two systems, featuring either an indirect activation by two layer DSD cascades (strategy 1) and a direct one layer DSD (strategy 2). Strategy 1 encompasses the critical step of a dual-sided invasion to eject longer ssDNA output strands via ATP-powered ligation for downstream DSD. In contrast to our earlier report[42], this enables a higher thermodynamic push and higher programmability in ATP-driven and transient DSD cascades, which is a new principle of general importance in

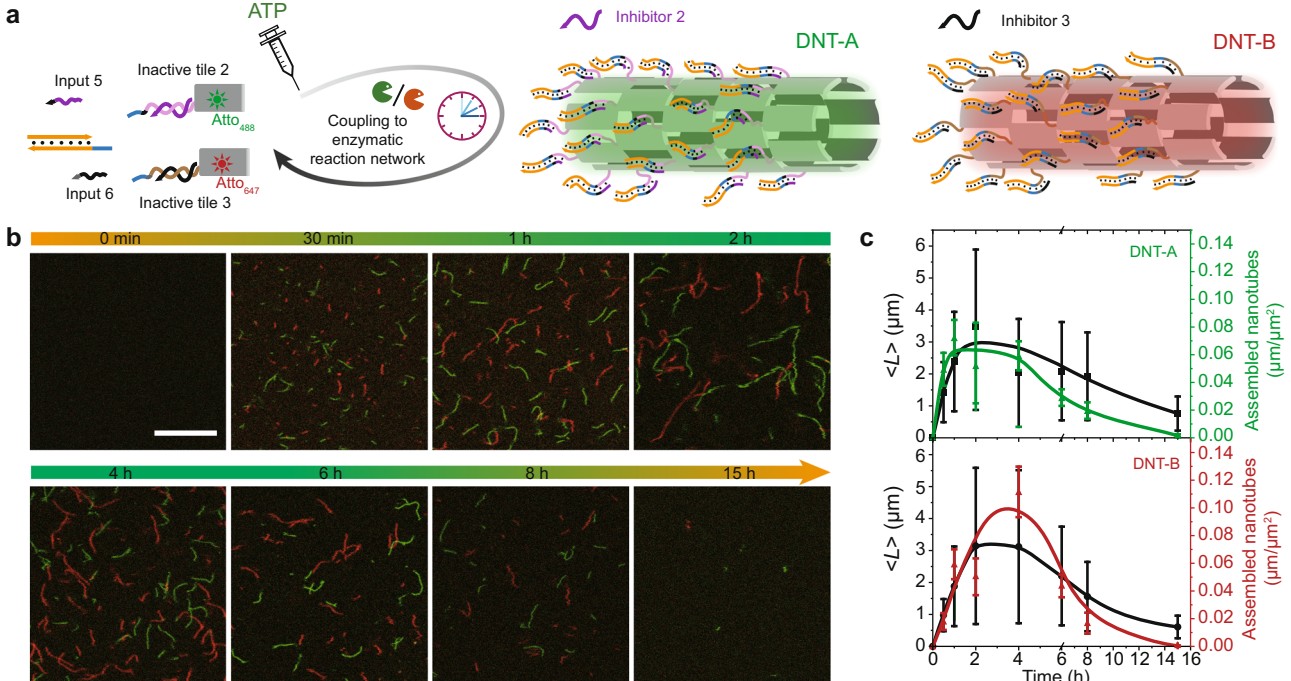

**Fig. 7 ATP-fueled transient self-sorting DNTs by orthogonal direct tile activation (Strategy 2). a** Schematic representation of ATP-fueled, transiently self-sorting DNT self-assemblies. **b** Time-dependent CLSM measurements of ATP-fueled transient self-sorting DNTs. Scale bar: 10 μm. **c** Time-dependent < L > s and yields of the both DNTs measured from CLSM images. Error bars correspond to the standard deviation of at least three random fields of view each with an area of 185.4 × 185.4 μm$^2$ from duplicate experiments. Conditions: 25 °C, 1 μM Inactive tile 2 (with 1 μM Inhibitor 2), 1 μM Inactive tile 3 (with 1 μM Inhibitor 3), 10 μM Complex 1, 5 μM Input 5, 5 μM Input 6, 0.8 WU μL$^{-1}$ T4 DNA ligase, 1.5 U μL$^{-1}$ BsaI, and 16 μM ATP.

concatenating these two CRNS. By reducing the two layer DSD cascades to a one layer DSD in strategy 2, we could show temporal tile activation for non-equilibrium self-assembly by direct ATP-fueled transient DSD on the assembling tile. By using ATP-fueled one layer DSD reaction network, significantly less DNA species are required for autonomous control of DNA nanostructure self-assembly. The DNA nanostructure self-assembly responds more swiftly to ATP-powered ligation, but the overall length of the structures is shorter due to direct deactivation of the building blocks. In future, it will be relevant to study more detailed the nanostructural differences in dynamics of both approaches. Furthermore, programming orthogonal molecular recognition for two DNA nanostructure self-assemblies allows to achieve ATP-fueled multicomponent self-sorting on a systems level. This can in principle be done for both approaches, but we realized this only for strategy 2 as clear proof of principle. This dynamic and transient self-sorting realizes artificial fuel-driven hierarchical non-equilibrium systems with both high system complexity and structural complexity, reaching a step closer to mimic hierarchical and sorted non-equilibrium systems in nature.

In the present study, we refined our previously reported ATP-fueled DSD to provide longer output sequences and a higher thermodynamic push[42], and further showed its application for autonomous control of DNA nanostructures assembly. The introduced concept to couple ATP dissipation to DNA nanostructure formation via intermediate layers is a generic concept that can be quite simply adapted to other DNA nanostructures, automatons or DNA computing in general. Looking to the future, we foresee that this ATP-fueled hierarchical non-equilibrium self-assembling system provides some insights for the fields of life-like materials, systems chemistry, and synthetic biology, and helps to better understand some principles of the dynamic structures in living systems. We envision, in particular, that the fuel-driven multicomponent DNA nanotubes may serve as artificial

cytoskeletal network for constructing synthetic cells with potential applications in biomedical engineering and nanofabrication.

## Methods

**DNA annealing**. All DNA strands were used as received. All the sequences were shown in Supplementary Table 1. The DNA strands intended for dsDNA complexes and substrates were dissolved in certain amounts of annealing buffer (10 mM Tris-HCl (pH 8.0), 50 mM NaCl) to make stock solutions with a concentration of ca. 1 mM calculated by the amounts of DNA given by IDT and the real concentrations were calculated via a UV–VIS spectroscopy. The dsDNA tiles were then annealed from two complementary ssDNA with the same stoichiometry at room temperature overnight. For comparison, Substrates 3 and 5 were annealed with both one and two equivalent amounts of its quencher strand. All the other DNA strands were dissolved in Milli-Q water with a concentration of ca. 0.5 mM as specified by IDT and the real concentrations were calculated via a UV–VIS spectroscopy. Inactive tile 1 was annealed in Tris acetate-edetate disodium (EDTA) (TAE)/Mg$^{2+}$ (40 mM Tris, 20 mM acetic acid, 1 mM EDTA, 12.4 mM MgCl$_2$, pH 8.0) containing 5 μM of all tile strands (S 1, S 2, S 3, S 4, and S 5), and 10 μM of Inhibitor 1 with a final volume of 50 μL. The final solution was annealed via a thermocycler by heating to 90 °C, and cooling to 20 °C (by decreasing 1 °C every 5 min) at a constant rate over a 6 h period to form inactive tiles with a concentration of 5 μM. For Inactive tile 2, S 2 was substituted by S 2* and 5 μM Inhibitor 2 was used. Afterwards, the solution (50 μL) was annealed by the same temperature ramp described above. The stock solution of Inactive tile 3 contains 5 μM of all tile strands (S 1-#2, S 2-#2, S 3-#2, S 4-#2, and S 5-#2)), and 5 μM of Inhibitor 3 in the TAE/Mg$^{2+}$ buffer, which was annealed by the same temperature described above. All the annealed stock solutions were stored at −20 °C for further use. All the experiments were performed by diluting the stock solutions and 10× NEB CutSmart buffer to Milli-Q water.

**ATP-fueled transient DNA strand displacement and its quantification by agarose gel electrophoresis**. The transient DSD experiments were performed in 1× NEB CutSmart buffer at 25 °C containing 20 μM Complex 1, 5 μM Substrate 1, 10 μM Input 1, 10 μM Input 2, 0.8 WU μL$^{-1}$ T4 DNA ligase, 1.5 U μL$^{-1}$ BsaI, and 40 μM ATP, with a total volume of 25 μL. At different time intervals (0 min, 10 min, 1 h, 2 h, 4 h, 6 h, 8 h, 10 h, 12 h, and 24 h), 2 μL aliquots of the reaction solution were collected and quenched by 3 μL of quenching buffer (200 mM EDTA, 10 mM Tris-HCl (pH 8.0), 50 mM NaCl). All the collected samples were stored at −20 °C for further analysis. Afterwards, 1 μL 6× DNA gel loading dye (10 mM Tris-HCl (pH 7.6) 0.03% bromophenol blue, 0.03% xylene cyanol FF, 60% glycerol 60 mM EDTA) was added to all the quenched samples and then the mixtures were

analyzed by 3 wt. % AGE containing ca. 0.02% (v/v) RotiR-GelStain at 80 V for 2.5 h run time at room temperature in TAE buffer (40 mM Tris, 20 mM acetic acid, 1 mM EDTA). The results were recorded by an INTAS *CHEMOSTAR touch* fluorescence imager via UV excitation. The ratio of DSD was calculated as the yield of Complex 2, and it was quantified by ImageJ software. Briefly, rectangular selections were made for every gel band of Complex 2 and the integral grayscale values for each gel band were then measured and calculated[40]. Furthermore, the grayscale value for Complex 2 at each time point was divided by the summed grayscale value of all gel bands at the corresponding time point to give an overall ratio of Complex 2. Since two equivalent amounts of Complex 1 were added, the final yield of Complex 2 can be calculated by Eq. (1) (see results in Supplementary Fig. 1a).

$$Yield = \frac{R1}{\left(\frac{n(Complex1)}{2n(Complex1+Substrate1)} + R1\right)} \tag{1}$$

In Eq. (1), $R1$ represents the overall grayscale value ratio of Complex 2 and $n$ represents the number of nucleobases in the dsDNA.

**ATP-fueled transient DSD monitored by allosteric switch of output strand via FRET**. The experiments were performed in 1× *NEB* CutSmart buffer at 25 °C containing 20 μM Complex 1, 5 μM Substrate 2, 10 μM Input 1, 10 μM Input 2, 0.8 WU μL$^{-1}$ T4 DNA ligase, 1.5 U μL$^{-1}$ BsaI, and varied concentrations of ATP (40, 60, and 80 μM). The experiments were carried out in a total volume of 25 μL in a black 384-well plate, and results were recorded after every 2 min by a plate reader with an excitation at 630 nm and an emission at 665 nm (10 nm for both band-widths). Note: a layer of hexadecane (8 μL) was added on top of the reaction solution to prevent any evaporation at 25 °C. Control experiments without Input 1 or Input 2 fueled by 40 μM ATP were performed at the same conditions described above.

**ATP-fueled transient DSD cascades**. The experiments were performed in 1× *NEB* CutSmart buffer at 25 °C containing 20 μM Complex 1, 5 μM Substrate 3, 1 μM Substrate 3 (with 2 equivalent amount of quencher strand), 10 μM Input 1, 5 or 10 μM Input 2, 0.8 WU μL$^{-1}$ T4 DNA ligase, 1.5 U μL$^{-1}$ BsaI, and 40 μM ATP. The experiments were carried out in a total volume of 25 μL in a black 384-well plate sealed by 8 μL hexadecane, and results were recorded after every 2 min by a plate reader (see above). For comparison, control experiments by using 1 μM Substrate 3 with only one equivalent amount of quencher strand and 10 μM Input 2 were also performed. The results were recorded by a plate reader the same as above.

**Repeated fueling of the ATP-fueled DSD cascades**. The experiments were performed in 1× *NEB* CutSmart buffer at 25 °C containing 20 μM Complex 1, 5 μM Substrate 1, 1 μM Substrate 3 (with two equivalent amount of quencher strand), 10 μM Input 1, 10 μM Input 2, 0.8 WU μL$^{-1}$ T4 DNA ligase, and 1.5 U μL$^{-1}$ BsaI, and initiated by adding 20 μM ATP. After the first transient lifecycle was finished, another 20 μM ATP was added to initiate the next transient lifecycle. This process was repeated for three times to give four transient lifecycles of the ATP-fueled transient DSD cascades. The initial experiments were carried out in a total volume of 25 μL in a black 384-well plate sealed by 8 μL hexadecane, and results were recorded after every 2 min by a plate reader (see above). Note: in order to not significantly dilute the DNA species in the well, 0.5 μL of 1 mM ATP solution in 1× *NEB* CutSmart buffer was added for each transient lifecycle.

**ATP-fueled DSD cascades at 37 °C**. The experiments were performed in 1× *NEB* CutSmart buffer at 37 °C containing 20 μM Complex 1, 5 μM Substrate 4, 1 μM Substrate 5 (with one or two equivalent amount of quencher strand), 10 μM Input 3, 10 μM Input 4, 0.8 WU μL$^{-1}$ T4 DNA ligase, 1.5 U μL$^{-1}$ BsaI, and varied concentration of ATP (40, 60, and 80 μM). The experiments were carried out in a total volume of 25 μL in a black 384-well plate sealed by 8 μL hexadecane, and results were recorded after every 2 min by a plate reader (see above).

**DNT assembly and disassembly by Activator and Deactivator strands**. Overall, the experiments were performed in 1× *NEB* CutSmart buffer at 25 °C. The stock solution of Inactive tile 1 (5 μM) was diluted 1:10 with 10× *NEB* CutSmart buffer and Milli-Q water to a final concentration of Inactive tile 1 of 0.5 μM and of Inhibitor 1 of 1 μM in 1× *NEB* CutSmart buffer (20 μL). The sample before activation was first imaged with CLSM using a 1:10 diluted solution in 1× *NEB* CutSmart buffer. Afterwards, the Activator strand (= Output 1) was added to the above solution (0.5 μM tile + 1 μM Inhibitor) at a final concentration of 3 μM. After reacting for 30 min, the sample was imaged with CLSM using a 1:10 diluted solution in 1× *NEB* CutSmart buffer to obtain the confocal images of assembled DNA nanotubes. Then, the Deactivator strand was added to the activated solution at a final concentration of 5 μM. After reacting for another 30 min, the sample was imaged with CLSM using 1:10 diluted solution in 1× *NEB* CutSmart buffer to verify the disassembled state.

**Stepwise control of DNT assembly and disassembly by sequential ATP-powered ligation and BsaI restriction**. First, the ATP-powered ligation-induced

DNT assembly was carried out in 1× *NEB* CutSmart buffer at 25 °C containing 0.5 μM Inactive tile 1 (0.5 μM tile + 1 μM Inhibitor), 20 μM Complex 1, 5 μM Substrate 1, 10 μM Input 1, 10 μM Input 2, 0.8 WU μL$^{-1}$ T4 DNA ligase, and 60 μM ATP in a total volume of 20 μL. After reacting for 2 h, the sample was imaged with CLSM using a 1:10 diluted solution in 1× *NEB* CutSmart buffer to obtain the confocal images of assembled DNTs. Afterwards, BsaI was added to the above reaction solution at a final concentration of 1.5 U μL$^{-1}$. After reacting for one day, the final sample was further imaged via CLSM using 1:10 diluted solution in 1× *NEB* CutSmart buffer to verify DNT degradation.

**Autonomous DNT controlled by ATP-fueled transient DSD cascades**. The experiments were performed in 1× *NEB* CutSmart buffer at 25 °C containing 0.5 μM Inactive tile 1 (0.5 μM tile + 1 μM Inhibitor), 20 μM Complex 1, 5 μM Substrate 1, 10 μM Input 1, 10 μM Input 2, 0.8 WU μL$^{-1}$ T4 DNA ligase, 1.5 U μL$^{-1}$ BsaI, and 40 μM ATP. The experiments were carried out in a total volume of 25 μL. At different time intervals (0 min, 30 min, 1 h, 2 h, 4 h, 6 h, 8 h, 12 h, and 24 h), 1 μL aliquots of the reaction solution were collected and diluted by 9 μL of 1× *NEB* CutSmart buffer. All the samples were images by CLSM upon collection. After 24 h, another batch of ATP was added to the remaining reaction solution at a final concentration of 40 μM. At different time intervals (25 h, 28 h, 36 h, and 48 h), 1 μL aliquots of the reaction solution were collected and diluted by 9 μL of 1× *NEB* CutSmart buffer. All the samples were images by CLSM upon collection.

**Direct ATP-fueled DSD on assembling tile for transient DNT self-assembly**. The experiments were performed in 1× *NEB* CutSmart buffer at 25 °C containing 1 μM Inactive tile 2 (1 μM tile + 1 μM Inhibitor), 5 μM Complex 1, 5 μM Input 5, 0.8 WU μL$^{-1}$ T4 DNA ligase, 1.5 U μL$^{-1}$ BsaI, and 8 μM ATP. The experiments were carried out in a total volume of 25 μL. At different time intervals (0 min, 30 min, 1 h, 2 h, 4 h, and 6 h), 1 μL aliquots of the reaction solution were collected and diluted by 9 μL of 1× *NEB* CutSmart buffer. All the samples were images by CLSM upon collection.

**ATP-fueled transient self-sorting DNTs**. The experiments were performed in 1× *NEB* CutSmart buffer at 25 °C containing 1 μM Inactive tile 2 (1 μM tile + 1 μM Inhibitor), 1 μM Inactive tile 3 (1 μM tile + 1 μM Inhibitor) 10 μM Complex 1, 5 μM Input 5, 5 μM Input 6, 0.8 WU μL$^{-1}$ T4 DNA ligase, 1.5 U μL$^{-1}$ BsaI, and 16 μM ATP. The experiments were carried out in a total volume of 25 μL. At different time intervals (0 min, 30 min, 1 h, 2 h, 4 h, 6 h, 8 h, 15 h), 1 μL aliquots of the reaction solution were collected and diluted by 9 μL of 1× *NEB* CutSmart buffer. All the samples were images by CLSM upon collection.

**Quantification of DNA nanotube**. We characterized the DNA nanotube by its average length and relative yield. We manually measured the length of each nanotube in multiple CLSM images from duplicate experiments via ImageJ using freehand line tool and measure length function. Then the sum of the nanotube length was divided by the number of measured nanotubes to give an average nanotube length. Simultaneously, the total length of the DNA nanotubes for each individual CLSM image was calculated, and the total nanotube length was divided by the image area to count the DNA nanotube density (μm per μm$^2$), which gives a relative yield of the DNA nanotube. DNA nanotubes from at least three random fields of view each with an area of 185.4 × 185.4 μm$^2$ were calculated.

**Determination of the lifetime for ATP-fueled transient DSD via FRET**. An illustration for lifetime calculation of the ATP-fueled transient DSD is shown in Supplementary Fig. 1b, where the lifetime for transient DSD was calculated from the point showing 80% recovery of the maximum decreased FI (FI$_{80\%\ recovery}$). Due to dye bleaching, the maximum decreased FI for the ATP-fueled system was calibrated by the FI of the system without ATP and 2.5% decrease of final FI was applied. For instance, the lifetime for the system fueled by 40 μM ATP was calculated as the point where FI$_{80\%\ recovery} = ((1-\text{FI}_{40}/\text{FI}_0)*0.8 + \text{FI}_{40}/\text{FI}_0)*97.5\%$, where FI$_{40}$ represents the normalized FI at the DySS.

## Data availability
The data that support the findings of this study are available from the corresponding author upon reasonable request. Source data for main Figs. 2d, e, 3b–f, 5d–g, 6c, d, 7c and Supplementary Figs. 1a, c, 2b, 5, 8b, 9b are provided with this paper. Source data are provided with this paper.

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

## Acknowledgements
We acknowledge support by the European Research Council starting Grant (Time-ProSAMat) Agreement 677960.

## Author contributions
J.D. and A.W. conceived the project. J.D. designed and conducted the experiments. J.D. analyzed the data and discussed the data with A.W., J.D. prepared the original draft. J.D. and A.W. reviewed and edited the manuscript. A.W. supervised the project.

## Funding

## Competing interests
The authors declare no competing interests.
