## [Peer Review File · Nature Communications]

Reviewers' Comments:

Reviewer #1:

Remarks to the Author:

The manuscript by Deng and Walther demonstrates the interconnection of an enzymatically powered strand displacement reaction requiring ATP with a process of DNA nanotube self-assembly. The DSD reactions (relevant DNA domains) are balanced to revert to their starting equilibrium point once ATP runs out and enzymes (ligase/restriction enzyme) stop processing their substrates. This generates a pulse in the output concentration whose duration can be tuned, primarily by changing the amount of ATP provided. This "pulsed" output is used to drive DNA nanotube assembly indirectly (strategy 1, output is released from complexes not participating in the tile that are a substrate for enzymes) or directly (strategy 2, tile toehold domains are a substrate for enzymes).

Both the DSD reaction and the DNA nanotube system were previously characterized in the literature, and this study identifies suitable modifications to existing designs so that their operation is compatible. The results are supported by data. Experiments include agarose gel electrophoresis and fluorescence spectroscopy to verify the transient ATP-fueled DSD operation, and fluorescence microscopy to measure nanotube formation.

My opinion is that the results are sound and of interest to the DNA nanotech/nanomaterials community, but better suited for a specialized journal. The novelty here is limited to the repurposing of the ATP-fueled DSD network demonstrated by the authors in [42] to release outputs to direct nanotube assembly, which was previously demonstrated by others.

Here are some comments to improve the manuscript:

- The first section of the paper can be shortened, in particular the paragraphs introducing the different strategies - see also next point.

- The figure schematics are often repetitive and not particularly informative.

For example, I suggest simplifying drastically fig. 1, which includes too many details; one ends up losing track of the main message.

Rather, more detail should be included about how the domains are designed in Figs 2a and 3a; I also think these two figures could be merged, since it seems the only difference is the addition of a reporter fluorophore/quencher pair in Fig. 3 (substrate 3). Note that the second part ("conditions") of the caption of Fig 2 must be wrong, as it refers to panels that do not show experiments.

In the following figures the DSD reaction is reproduced every time, while it would be sufficient to replace with a box and report only new complexes/outputs that are needed for binding to the nanotubes.

- While the authors acknowledge openly that the ATP-fueled DSD network is a modification of their previous work [42], the way they reference papers [35, 36] and [44,45] is inaccurate and misleading. DAE-E tiles were first described in [44,45]. DAE-E based DNA nanotubes were first demonstrated by Rothmund in <https://doi.org/10.1021/ja044319l> (not cited).

Autonomous dynamic assembly of DAE-E DNA nanotubes was introduced in refs [35,36], where tiles were engineered to be activated/deactivated via strand displacement powered by enzymes producing and degrading RNA. The idea to control nanotubes described here is identical to the designs of invader/anti-invader in [36] — in fact, I would recommend preserving that nomenclature since the reactions are essentially the same with minor adaptation. Even the nanotube schematics in this paper are directly taken from [36], to the point of being almost edited pdfs from Fig. 1 of [36]. While it is perfectly fine to adapt designs & figures from the literature, it should be acknowledged clearly to avoid unpleasant suspicions of plagiarism.

- The property of DNA tiles to self-sort has nothing to do with the upstream DSD reaction. It is rather an intrinsic property of tiles designed to be orthogonal in sequence; they would self-sort regardless of the dynamic control. So while I can see the last section is an interesting demonstration, I fail to see the intellectual contribution/novelty. Also similar demonstrations of transient assembly on multi-nanotube populations are in [34,36] (with transcription networks instead of ATP/DSD).

- Please clarify the technical difference between the “equivalent” quencher strands described at the top of page 6. There is a hint to a difference in length/possibly sequence, but this is not explained.
- Error bars for nanotube length: is this the standard deviation of the mean length measured in different experiments? Or of the length histogram?
- SI figs 6 and 8 - please increase contrast/brightness, nanotubes are barely visible.

Reviewer #2:

Remarks to the Author:

In this work the authors described a new strategy to achieve an autonomous ATP-fueled DNA nanotube assembly, controlled by DNA strand displacement (DSD) reactions that are regulated by an enzymatic reaction network of concurrent ATP-powered ligation and restriction of DNA components.

The strategy employed to obtain the assembly and disassembly of DNA nanotubes has been previously demonstrated by Franco and co-workers (Ref 36). While here, the authors provide a new advance, introducing an ATP-powered enzymatic reaction network to induce the transient assembly.

The authors first described a new strategy to achieve an ATP-fueled transient DSD and an ATP-fueled transient DSD cascades. After that, they demonstrated two different approaches for the autonomous DNT assembly: 1) an indirect control of DNT assembly by ATP-fueled DSD cascades and 2) a direct control of DNT assembly by ATP-fueled transient ligation and ensuing DSD on the assembling tiles. They further demonstrated ATP-fueled and transiently self-sorting DNTs by programming orthogonal molecular recognition into parallel systems.

Comments:

1. Here, the authors extend their previously reported strategy to achieve a “Fuel-driven transient DNA strand displacement” (Deng, Walther. “Fuel-Driven Transient DNA Strand Displacement Circuitry with Self-Resetting Function” J. Am. Chem. Soc. 142, 21102). Can they explain the advantages of this new strategy?

2. The authors wrote (page 10) “we re-engineered a double-crossover DNA tile known as DAE-E”.

The DAE-E tiles used in this work are those reported by Franco and co-workers: Green L. N., et al. “Autonomous dynamic control of DNA nanostructure self-assembly.” Nat. Chem. 11, 510-520 (2019). The authors should probably clarify this in the text.

3. The authors should correct the “fig 4c,g” with “fig. 5c,g” (page 11).

4. In fig. 5g the authors described two transient cycles through the average length of DNA nanotubes and through the yield of assembly reported as “assembled nanotubes” ($\mu\text{m}/\mu\text{m}^2$). It is not clear to this reviewer how this parameter has been calculated.

5. The authors showed (fig 2d,e and fig 3e,f) the programmability of the ATP-fueled transient DSD by varying the concentration of the ATP. While, they did not describe the lifetime of nanotubes assembly at different ATP concentrations. Can they describe how the ATP concentration can influence the programmability of the transient nanotubes assembly?

Reviewer #3:

Remarks to the Author:

Recent years have witnessed tremendous progress over devolvement of synthetic dissipative materials with emergent functions. However, a close inspection of the relevant examples reveals

lack of the complexity natural dissipative system offers. In the present manuscript, Walther and coworkers presents DNA based multicomponent self-assembling nanotubes which is coupled to concatenated chemical reaction network. Based on their expertise of ATP-fueled ligation and cleavage of phosphodiester bond (enzymatic reaction network (ERN)), they have reported communication between ERN to realize ATP fueled DNA cascades. However, realization of higher order DNA nanostructures by multiple connected ERN remains elusive. Thus, the present manuscript successfully attempts ATP fueled concatenated reaction network for downstream hierarchical DNA nanostructures. It reports two strategies to couple ERN to regulate transient formation of DNA nanotubes. In the first strategy the output DNA strand of CRN1 induces a strand displacement reaction to initiate the DNA nanotube formation. In the second strategy the output strand of CNR1 couples to inactivated DNA tiles to liberate a free single stranded DNA to initiate the assembly. The second strategy was further extended to a multicomponent system to observe a self-sorted transient DNA nanotube using orthogonal molecular recognition of output strand of CRN1. Overall, the system represents design of complex CRN with hierarchical structures by concatenation of CRNs. The manuscript can be accepted in nature communication however some of the suggestions are as follows.

Major Points

- The design principle of the DNA nanotubes was inspired from the work by Franco (Nat. Chem. 11, 510-520 (2019)). In their previous manuscript the authors have reported hierarchical assembly of colloidal particles and self-sorting utilizing multivalency (Nat. Commun. 11, 3658 (2020)). The present system intends to investigate formation of higher order DNA nanostructures. Please highlight the novelty in a more appropriate manner.
- In figure 2d in absence of ATP there is a decrease in $FIEM(t)/FIEM(t=0)$ at 0 eqv of ATP over time. The authors should explain it.
- Can they comment about % conversion of (formation of complex 2) in presence of ATP. This will be very important to determine the efficiency of the system.
- How did the authors calculate the lifetime from figure 2d? Treatment of the data and procedure should be included in the SI.
- The authors have mentioned two equivalent of input 2 leads to a faster kinetics. Why particularly increase the equivalent of 2 than that of increase the concentration of input 1?
- The authors should mention the equivalent of quencher strand in Figure 3c for better readability of the Figure.
- In figure 3d why there is a damping in the first cycle after which it stays constant? With 80 % recovery in each cycle, I would expect a sequential damping.
- The authors have mentioned the average length of DNA nanotubes obtained from ATP powered DNA cascades are having lower size compared to the system directly activated by activator strand. Is it due to lower efficiency of the DNA cascade in releasing activator strand?
- The self-sorted transient DNA nanotubular self-assembly shown is quite remarkable. However, the narcissistic self-sorting occurs from difference in sticky ends of DNA tiles and independent of transient DNA cascade.

Minor Points

- The figure caption of 5d-e needs to be expanded for better understanding.
- I did not get the idea of calculation of "the average length of DNT per square micrometer". It will vary depending on where the author is looking under the microscope.
- How did the authors decided to image the samples at 2 h or 8 h to visualize the nanotubes? Did they perform any gel electrophoresis to find out at what position the output strand concentration is maximum? Or the kinetics of the dynamic cascade in presence of and absence DNA tiles follows similar trend?
- Why no refueling is performed in strategy 2 for transient DNA nanotube formation. Also, the authors have utilized different tile and enzyme concentration in strategy 2 compared to strategy 1. This makes direct comparison of strategy 1 and strategy 2 difficult. Any particular reason for it?
- Can they comment about how the nanotubes are getting disassembled? From the end or it can start from any position in the middle?
- The authors have mentioned the DNA nanotubes were imaged by diluting the solutions. What is the reason behind this?
- The ATP fueled DNA cascade shown at 37 °C has a very low efficiency.

Reviewer #1 (Remarks to the Author):

The manuscript by Deng and Walther demonstrates the interconnection of an enzymatically powered strand displacement reaction requiring ATP with a process of DNA nanotube self-assembly. The DSD reactions (relevant DNA domains) are balanced to revert to their starting equilibrium point once ATP runs out and enzymes (ligase/restriction enzyme) stop processing their substrates. This generates a pulse in the output concentration whose duration can be tuned, primarily by changing the amount of ATP provided. This “pulsed” output is used to drive DNA nanotube assembly indirectly (strategy 1, output is released from complexes not participating in the tile that are a substrate for enzymes) or directly (strategy 2, tile toehold domains are a substrate for enzymes).

Both the DSD reaction and the DNA nanotube system were previously characterized in the literature, and this study identifies suitable modifications to existing designs so that their operation is compatible. The results are supported by data. Experiments include agarose gel electrophoresis and fluorescence

spectroscopy to verify the transient ATP-fueled DSD operation, and fluorescence microscopy to measure nanotube formation.

My opinion is that the results are sound and of interest to the DNA nanotech/nanomaterials community, but better suited for a specialized journal. The novelty here is limited to the re-purposing of the ATP-fueled DSD network demonstrated by the authors in [42] to release outputs to direct nanotube assembly, which was previously demonstrated by others.

Response: In general, we appreciate the comments. However, we don't agree that the novelty of the present study is compromised by our previous report [42]. The main claim in the present manuscript is the autonomous control of downstream DNA nanostructures by hierarchical concatenation of several distinct chemical reaction networks (CRNs) upstream. This is still very rare in literature in general and had not at all been addressed in our previous report. In fact, recent years have witnessed great progress in non-equilibrium self-assemblies. However, most of them focus on single type of chemical reaction networks and their connections to self-assembling building blocks. In the present study, we report hierarchically concatenated CRNs to regulate self-assembling processes, which provides a higher level model system to biological systems, where non-equilibrium assemblies are controlled by hierarchically layered CRNs for various biofunctions.

Although we previously reported ATP-fueled DNA strand displacement (DSD) cascades [42], the concept in the present study is completely different: 1. There was absolutely no connection to any nanostructural level. 2. Moreover, to enable this connection, we also needed to rethink the previous ATP-fueled DSD system to a system with dual-side ATP-fueled invasion of the important double-stranded DNA complex, which, critically, results in a higher thermodynamic push and also a higher programmability of any downstream DSD cascades compared to our previous report. The ejected output strand is longer. These two key points are non-trivial and substantial extensions to the previously reported strategy, as we use it for new systems allowing to achieve higher system and structural complexities.

Critically, we believe our study is not only of great interest to the DNA community, but also to the researchers in the fields of systems chemistry and non-equilibrium self-assembly in general. Admittedly, DNA nanotube systems were previously characterized in the literature, but here it is only a relevant model system to prove our concept. It could be any other DSD regulated nanostructures. Moreover, ATP, as a biofuel, has not been investigated to control such a process before, which, however, may be a crucial aspect/advantage for further applications in a biological environment, as our system can directly use the biofuel available therein.

We have added more clarifications in our revised manuscript, which are also shown below:

Page 2-3: "However, autonomous dynamic control of high-order DNA nanostructures by multiple concatenated distinct CRNs is still rare. A relevant report is the autonomous control of DNA nanotube assembly via RNA transcription machinery by Franco and coworkers. This however uses sequential steps of reactions,³⁶ and proceeds rather energetically downhill from the beginning. Hence, we intended to develop ATP-fueled hierarchical self-assembly in an energetically up-hill fashion using cyclic reaction network, in which the ATP-fueled CRN module transduces via a different CRN to achieve a hierarchical

structure downstream. Success in this direction would help to better mimic the hierarchical non-equilibrium systems in nature and provide more insights into future life-like materials design.”

Page 4: “We build on our previous ATP-driven DSD circuits studied exclusively on a molecular level⁴², but made critical design improvements by designing a Substrate 1 that can be invaded from both sides via ATP-powered ligation (using Complex 1 and Input 1 and Input 2) to allow for a longer Output Regulator strand to be ejected from Substrate 1. This dual-side invasion to release longer Output strands provides a higher thermodynamic push and higher programmability of the downstream DSD cascades. The ligation transiently biases the energy landscape of the strand displacement circuit. Note that Input 1 and Input 2 must be relatively short sequences (with respect to the operating temperature) so that the DSD only happens when ligation powered by ATP proceeds.”

In discussion: “In the present study, we refined our previously reported ATP-fueled DSD to provide longer output sequences and a higher thermodynamic push,⁴² and further showed its application for autonomous control of DNA nanostructures assembly. The introduced concept to couple ATP dissipation to DNA nanostructure formation via intermediate layers is a generic concept that can be quite simply adapted to other DNA nanostructures, automatons or DNA computing in general..”

Here are some comments to improve the manuscript:

- The first section of the paper can be shortened, in particular the paragraphs introducing the different strategies - see also next point.

Response: We have revised our Fig. 1 and cut down the length of the first section accordingly.

- The figure schematics are often repetitive and not particularly informative. For example, I suggest simplifying drastically fig. 1, which includes too many details; one ends up losing track of the main message.

Response: Thanks for the suggestion. We have simplified Fig. 1 in our revised manuscript, which is now mainly focusing on different strategies to control the autonomous DNA nanotube assembly.

Rather, more detail should be included about how the domains are designed in Figs 2a and 3a; I also think these two figures could be merged, since it seems the only difference is the addition of a reporter fluorophore/quencher pair in Fig. 3 (substrate 3). Note that the second part (“conditions”) of the caption of Fig 2 must be wrong, as it refers to panels that do not show experiments.

Response: We have included more details of different domains in Fig. 2a and 3a in our revised manuscript. We decided to not merge them together as they show different approaches, one step DSD versus two step DSD cascades. Also, it is hard to concisely display all the data by a merged figure. The conditions in the caption have been redirected to specific experiments.

In the following figures the DSD reaction is reproduced every time, while it would be sufficient to replace with a box and report only new complexes/outputs that are needed for binding to the nanotubes.

Response: Thanks for the suggestion. We prefer to include the whole DSD reaction network for the nanotube assembly for better understanding of the whole autonomous process, because there is also a

released strand from BsaI restriction to deactivate the assembled nanotube. Also the DSD reactions in Fig. 4b and Fig. 5a are different. Simply replacing the DSD network with a box cannot make it better.

- While the authors are acknowledge openly that the ATP-fueled DSD network is a modification of their previous work [42], the way they reference papers [35, 36] and [44,45] is inaccurate and misleading. DAE-E tiles were first described in [44,45]. DAE-E based DNA nanotubes were first demonstrated by Rothmund in <https://doi.org/10.1021/ja044319l> (not cited).

Response: Thanks for the comment. We have corrected this in our revised manuscript, which is also shown below:

Page 9: “Those are conceptually derived from double-crossover DNA tile known as DAE-E first reported by Seeman and coworkers,^{44, 45} whereas Rothmund and coworkers firstly reported assembly for DNA nanotubes.⁴⁶ Towards our first strategy, we adapted the reengineered tiles from Franco and Ricci groups to our ATP-fueled system.^{34, 36}”

Autonomous dynamic assembly of DAE-E DNA nanotubes was introduced in refs [35,36], where tiles were engineered to be activated/deactivated via strand displacement powered by enzymes producing and degrading RNA. The idea to control nanotubes described here is identical to the designs of invader/anti-invader in [36] — in fact, I would recommend preserving that nomenclature since the reactions are essentially the same with minor adaptation. Even the nanotube schematics in this paper are directly taken from [36], to the point of being almost edited pdfs from Fig. 1 of [36]. While it is perfectly fine to adapt designs & figures from the literature, it should be acknowledged clearly to avoid unpleasant suspicions of plagiarism.

Response: The reviewer is not correct in stating that these strategies are identical. There are important conceptual differences on the design of the reaction network, the thermodynamics and in the experimental realization.

For the conceptual difference on the reaction network level and thermodynamics we refer to the explanation from above: Page 2-3: “However, autonomous dynamic control of high-order DNA nanostructures by multiple concatenated distinct CRNs is still rare. A relevant report is the autonomous control of DNA nanotube assembly via RNA transcription machinery by Franco and coworkers. This however uses **sequential steps of reactions,³⁶ and proceeds rather energetically downhill from the beginning.** Hence, we intended to develop ATP-fueled hierarchical self-assembly in an energetically **up-hill fashion using cyclic reaction network**, in which the ATP-fueled CRN module transduces via a different CRN to achieve a hierarchical structure downstream. Success in this direction would help to better mimic the hierarchical non-equilibrium systems in nature and provide more insights into future life-like materials design.”

In ref. 36, the authors reported transient degradation of DNA nanotube, where the DNA nanotube is the starting state. Hence, invader and anti-invader can better describe their system. However, we start with inhibited tiles, and the released activator can trigger nanotube assembly, followed by deactivation via deactivator. We are very hesitant to name the activator as anti-invader in our study, which does not seem to be logically correct as we don't have pre-assembled structure being invaded beforehand. The design of the nanotube in our figure certainly relates to ref.36, but it is not copy/paste. We have acknowledged this in our revised manuscript, which is also shown as below:

“The design of the double-crossover tile and DNA nanotube is adapted from a previous report.^{36,}”

- The property of DNA tiles to self-sort has nothing to do with the upstream DSD reaction. It is rather an intrinsic property of tiles designed to be orthogonal in sequence; they would self-sort regardless of the dynamic control. So while I can see the last section is an interesting demonstration, I fail to see the intellectual contribution/novelty. Also similar demonstrations of transient assembly on multi-nanotube populations are in [34,36] (with transcription networks instead of ATP/DSD).

Response: Although multi-nanotube assembly is an intrinsic property of the sequence design in the tiles, we aim to achieve autonomous control of multicomponent self-assembly via a single enzymatic reaction network. Owing to the excellent programmability of DNA sequences in ATP-fueled transient dynamic ligation, different ATP-fueled DSD systems can be controlled by a single enzymatic reaction network, leading to the control of multi-nanotube assembly.

In fact, we show how to control multiple independent self-assembly processes by a single initial toolbox, and they share the same lifetime as regulated by ATP concentration. Let us phrase it differently, if we were to ask “how to self-sort DNA nanotubes transiently using ATP as a fuel (without the data in the present manuscript)?”, it would probably not be straightforward to come up with an idea. Indeed, the reviewer is right, this part works as an extension and demonstration of our system. We believe that we do not claim more than what it really is.

- Please clarify the technical difference between the “equivalent” quencher strands described at the top of page 6. There is a hint to a difference in length/possibly sequence, but this is not explained. Response: We have modified the strands in Figure 3a to show this difference. We also have the clarification in our manuscript, which is also shown as below:

“Critically, due to the interference of the regeneration of Substrate 3 by the hybridization between Output 3 and the long ssDNA dissociated from Intermediate 2, two equivalents of the quencher strand are needed for swift Substrate 3 regeneration. By using only one equivalent of the quencher strand, the FI recovery is quite time-consuming (Fig. 3c vs. Fig 3b), because the quencher strand has only one more nucleobase to compete with the long ssDNA from Intermediate 2 to hybridize with Output 3.”

- Error bars for nanotube length: is this the standard deviation of the mean length measured in different experiments? Or of the length histogram?

Response: Error bars correspond to the standard deviation of at least three random fields of view from duplicate experiments, while the data in length histogram is also analyzed from duplicate experiments.

- SI figs 6 and 8 - please increase contrast/brightness, nanotubes are barely visible. Response: We have adjusted it in our revised manuscript.

Thank you for detailed comments to improve clarity.

Reviewer #2 (Remarks to the Author):

In this work the authors described a new strategy to achieve an autonomous ATP-fueled DNA nanotube assembly, controlled by DNA strand displacement (DSD) reactions that are regulated by an enzymatic reaction network of concurrent ATP-powered ligation and restriction of DNA components. The strategy employed to obtain the assembly and disassembly of DNA nanotubes has been previously demonstrated by Franco and co-workers (Ref 36). While here, the authors provide a new advance, introducing an ATP-powered enzymatic reaction network to induce the transient assembly.

The authors first described a new strategy to achieve an ATP-fueled transient DSD and an ATP-fueled transient DSD cascades. After that, they demonstrated two different approaches for the autonomous DNT assembly: 1) an indirect control of DNT assembly by ATP-fueled DSD cascades and 2) a direct control of DNT assembly by ATP-fueled transient ligation and ensuing DSD on the assembling tiles. They further demonstrated ATP-fueled and transiently self-sorting DNTs by programming orthogonal molecular recognition into parallel systems.

Comments:

1. Here, the authors extend their previously reported strategy to achieve a “Fuel-driven transient DNA strand displacement” (Deng, Walther. “Fuel-Driven Transient DNA Strand Displacement Circuitry with Self-Resetting Function” J. Am. Chem. Soc. 142, 21102).

Can they explain the advantages of this new strategy?

Response: Please see also comment 1 of reviewer 1:

We added additionally information for clarification on

Page 4: “We build on our previous ATP-driven DSD circuits studied exclusively on a molecular level⁴², but made critical design improvements by designing a Substrate 1 that can be invaded from both sides via ATP-powered ligation (using Complex 1 and Input 1 and Input 2) to allow for a longer Output Regulator strand to be ejected from Substrate 1. This dual-side invasion to release longer Output strands provides a higher thermodynamic push and higher programmability of the downstream DSD cascades. The ligation transiently biases the energy landscape of the strand displacement circuit. Note that Input 1 and Input 2 must be relatively short sequences (with respect to the operating temperature) so that the DSD only happens when ligation powered by ATP proceeds.”

2. The authors wrote (page 10) “we re-engineered a double-crossover DNA tile known as DAE-E”. The

DAE-E tiles used in this work are those reported by Franco and co-workers: Green L. N., et al.

“Autonomous dynamic control of DNA nanostructure self-assembly.” Nat. Chem. 11, 510-520 (2019). The authors should probably clarify this in the text.

Response: Thanks for the comments. We have clarified this in our revised manuscript, which is also shown below:

Page 9: “Those are conceptually derived from double-crossover DNA tile known as DAE-E first reported by Seeman and coworkers,^{44, 45} whereas Rothmund and coworkers firstly reported assembly for DNA nanotubes.⁴⁶ Towards our first strategy, we adapted the reengineered tiles from Franco and Ricci groups to our ATP-fueled system.^{34, 36}”

3. The authors should correct the “fig 4c,g” with “fig. 5c,g” (page 11).

Response: This has been corrected in our revised manuscript.

4. In fig. 5g the authors described two transient cycles through the average length of DNA nanotubes and through the yield of assembly reported as “assembled nanotubes” ($\mu\text{m}/\mu\text{m}^2$). It is not clear to this reviewer how this parameter has been calculated.

Response: Thank you, we added details to the SI:

Page S7 and also here: “We characterized the DNA nanotube by its average length and relative yield. We manually measured the length of each nanotube in multiple CLSM images from duplicate experiments via ImageJ using freehand line tool and measure length function. Then the sum of the nanotube length was divided by the number of measured nanotubes to give an averaged nanotube length. Simultaneously, the total length of the DNA nanotubes for each individual CLSM image was calculated, and the total nanotube length was divided by the image area to count the DNA nanotube density ($\mu\text{m}/\mu\text{m}^2$), which gives a relative yield of the DNA nanotube. DNA nanotubes from at least three random fields of view with an area of $185.4 \times 185.4 \text{ nm}^2$ were calculated.

5. The authors showed (fig 2d,e and fig 3e,f) the programmability of the ATP-fueled transient DSD by varying the concentration of the ATP. While, they did not describe the lifetime of nanotubes assembly at different ATP concentrations. Can they describe how the ATP concentration can influence the programmability of the transient nanotubes assembly?

Response: We made additional comments on page 10/11. We have shown in several reports before that this ATP-driven network has a close to linear correspondence to the ATP concentration regarding lifetime control. It would be redundant to reiterate this here and also loose a bit focus here as the focus in this manuscript is on the connectivity of two CRNs and the nanostructures.

We added the following on page 10/11

“It is worth noting that the lifetime for the DNA nanotube can in principle be programmed by ATP concentration due to its direct connection to the ATP-dissipative cyclic ERNs.⁴⁰ In this study, we simply fueled the system with sufficient ATP (2 times of needed ATP for full ligation) to ensure a DySS of the DNA nanotube assembly. The system cannot enter into its DySS when there is not enough ATP available for the ligation.⁴⁰”

Thank you for detailed comments to improve clarity.

Reviewer #3 (Remarks to the Author):

Recent years have witnessed tremendous progress over development of synthetic dissipative materials with emergent functions. However, a close inspection of the relevant examples reveals lack of the complexity natural dissipative system offers. In the present manuscript, Walther and coworkers presents DNA based multicomponent self-assembling nanotubes which is coupled to concatenated chemical reaction network. Based on their expertise of ATP-fueled ligation and cleavage of phosphodiester bond (enzymatic reaction network (ERN)), they have reported communication between ERN to realize ATP fueled DNA cascades. However, realization of higher order DNA nanostructures by multiple connected ERN remains elusive. Thus, the present manuscript successfully attempts ATP fueled concatenated reaction network for downstream hierarchical DNA nanostructures. It reports two strategies to couple ERN to regulate transient formation of DNA nanotubes. In the first strategy the output DNA strand of CRN1 induces a strand displacement reaction to initiate the DNA nanotube formation. In the second strategy the output strand of CNR1 couples to inactivated DNA tiles to liberate a free single stranded DNA to initiate the assembly. The second strategy was further extended to a multicomponent system to observe a self-sorted transient DNA nanotube using orthogonal molecular recognition of output strand of CRN1. Overall, the system represents design of complex CRN with hierarchical structures by concatenation of CRNs. The manuscript can be accepted in nature communication however some of the suggestions are as follows.

Major Points

- The design principle of the DNA nanotubes was inspired from the work by Franco (Nat. Chem. 11, 510-520 (2019)). In their previous manuscript the authors have reported hierarchical assembly of colloidal particles and self-sorting utilizing multivalency (Nat. Commun. 11, 3658 (2020)). The present system intends to investigate formation of higher order DNA nanostructures. Please highlight the novelty in a more appropriate manner.

Response: Please see also our comments above for the difference to the work by Franco and co-workers. There are important conceptual differences on the design of the reaction network, the thermodynamics and in the experimental realization. The RNA transcription machinery for controlling DNA nanotube assembly by Franco (Nat. Chem. 11, 510-520 (2019)) uses sequential reactions in a linear fashion, which is rather energetically downhill from the beginning. In our study, we use cyclic reaction network, where the DSD cascades and downstream structure assembly are maintained in a non-equilibrium dynamic steady state by an up-hill formation of a dynamic covalent phosphodiester bond by ligation at the complementary sticky ends on the DNA tiles and simultaneous hydrolysis by BsaI for monomer regeneration.

Additionally, the study is a fundamentally different approach compared to transient multivalency (Nat. Commun. 11, 3658 (2020)). In the present study, we highlight the connection of higher order DNA

nanostructures assembly to hierarchically concatenated distinct chemical reaction networks: (1) ATP-powered ligation/restriction of DNA components and (2) input strand-mediated DNA strand displacement (DSD) using energy gains provided in DNA toeholds.

We have rephrased this in our revised manuscript in particular regarding Franco's work, which is also shown below:

We have added more clarifications in our revised manuscript, which are also shown below:

Page 2-3: "However, autonomous dynamic control of high-order DNA nanostructures by multiple concatenated distinct CRNs is still rare. A relevant report is the autonomous control of DNA nanotube assembly via RNA transcription machinery by Franco and coworkers. This however uses sequential steps of reactions,³⁶ and proceeds rather energetically downhill from the beginning. Hence, we intended to develop ATP-fueled hierarchical self-assembly in an energetically up-hill fashion using cyclic reaction network, in which the ATP-fueled CRN module transduces via a different CRN to achieve a hierarchical structure downstream. Success in this direction would help to better mimic the hierarchical non-equilibrium systems in nature and provide more insights into future life-like materials design."

- In figure 2d in absence of ATP there is a decrease in $FIEM(t)/FIEM(t=0)$ at 0 eqv of ATP over time. The authors should explain it.

Response: The decrease of FI is about 2.5 % after 24 h, which is attributed to dye bleaching during the measurement. We have mentioned this in our revised manuscript, which is also shown below:

"The slightly decreased FI (ca. 2.5 %) for the system without ATP is due to dye bleaching during the measurements."

- Can they comment about % conversion of (formation of complex 2) in presence of ATP. This will be very important to determine the efficiency of the system.

Response: The dynamic system shows a fraction of DSD of ca. 50 % at its DySS, as calculated by the yield of Complex 2 compared to Substrate 1 (Supplementary Fig. 1a). We do not achieve 100 % conversion of Complex 2, because our system is dynamic and includes forward and backward reaction (cyclic reaction network). In principle, the conversion can be increased by decreasing the BsaI concentration to limit the bond cleavage. We further clarified this in our revised manuscript, which is also shown below:

"It is worth noting that the fraction of DSD can be further tuned by the ratio of both enzymes⁴²."

- How did the authors calculate the lifetime from figure 2d? Treatment of the data and procedure should be included in the SI.

Response: The lifetime for the transient DSD increases from ca. 12 to 19 h, as calculated from the point showing 80 % recovery of the maximum decreased FI. We have added the procedure of the treatment of the data in our revised supporting information (SI Fig. 1).

- The authors have mentioned two equivalent of input 2 leads to a faster kinetics. Why particularly increase the equivalent of 2 than that of increase the concentration of input 1?

Response: Increasing input 1 and input 2 should have the same effect. Hence, we directly used 2 equivalents of input 1 and only adjusted the amount of input 2.

- The authors should mention the equivalent of quencher strand in Figure 3c for better readability of the Figure.

Response: We have added this in our revised Figure.

- In figure 3d why there is a damping in the first cycle after which it stays constant? With 80 % recovery in each cycle, I would expect a sequential damping.

Response: After ATP is consumed, the Output ssDNA is continuously depleted by hybridizing with its complementary strand, leading to quenched fluorescence. The damping is attributed to a gradually lower concentration of the Output strand for hybridizing with the quencher strand. So it is the case for every cycle. With longer measuring time, it still slowly decreases (Fig. 3b). We roughly have 90 % recovery, which is satisfactory. The reason we do not have 100 % recovery is because the quencher strand has only one more nucleobase to compete with the long ssDNA from Intermediate 2 to hybridize with Output 3. Hence, we used two equivalents of the quencher strand to facilitate this process.

- The authors have mentioned the average length of DNA nanotubes obtained from ATP powered DNA cascades are having lower size compared to the system directly activated by activator strand. Is it due to lower efficiency of the DNA cascade in releasing activator strand?

Response: We mentioned the direct strategy shows a shorter average length of nanotubes than the indirect strategy. This is not due to lower efficiency of the DSD cascades but rather related to the dynamics of the enzymatic reaction networks, where we have significantly more enzymes directly work on the DNA nanotubes in the direct strategy, meaning more frequent cutting of the nanotubes.

Page 12: "Compared to the system controlled by DSD cascades (above), shorter DNTs are observed at the DySS, which is attributed to the more dynamic nature of the DNTs due to direct restriction on the DNTs and also a higher ratio of BsaI to cleavable sequences (1.5 U, μL to 1 μM versus 1.5 U, μL to 10 μM). The higher DNT yield (as judged by the length per μm^2) is attributed to more efficient tile activation."

- The self-sorted transient DNA nanotubular self-assembly shown is quite remarkable. However, the narcissistic self-sorting occurs from difference in sticky ends of DNA tiles and independent of transient DNA cascade.

Response: The self-sorting of the nanotubes is inherently programmed by the sequences of the overhangs on the assembling tiles. However, these sorted nanotubes are connected to the same enzymatic reaction network. This further emphasizes the importance of the hierarchically concatenated distinct CRNs, which allows to control multiple independent self-assembling processes by a single initial toolbox, and they share the same lifetime as regulated by ATP concentration.

Minor Points

- The figure caption of 5d-e needs to be expanded for better understanding. Response: Thanks for the suggestion. We have added more details in our revised figure caption.

- I did not get the idea of calculation of “the average length of DNT per square micrometer”. It will vary depending on where the author is looking under the microscope.

Response: We showed zoomed-out pictures in supporting information that display good homogeneity. We counted random fields of view and the procedure is now further detailed in the SI (page S7)

- How did the authors decided to image the samples at 2 h or 8 h to visualize the nanotubes? Did they perform any gel electrophoresis to find out at what position the output strand concentration is maximum? Or the kinetics of the dynamic cascade in presence of and absence DNA tiles follows similar trend? Response: This refers to the FRET measurement in Fig. 3b, where we used the same concentration for ATP, both enzymes and ligation substrates, and the lifetime is ca. 13 h. Since the assembly of the nanotube is directly controlled by the DSD cascades, the lifetime for the DNA nanotube should be similar to the lifetime of the DSD cascades. We measured the nanotube at an interval of 2 h, while we did not show the CLSM images at all the time points in the figure.

- Why no refueling is performed in strategy 2 for transient DNA nanotube formation. Also, the authors have utilized different tile and enzyme concentration in strategy 2 compared to strategy 1. This makes direct comparison of strategy 1 and strategy 2 difficult. Any particular reason for it?

Response: Both strategies are controlled by the ATP-fueled enzymatic reaction network. We believe we have made it clear that the enzymatic system can be refueled for multiple cycles in Fig. 3 for the network and in Fig. 5 for the DNT assembly. The reliability is sufficiently shown and it does not add much to perform these repeated activations for all systems.

We did not change the enzyme concentration (0.8 WU/ μ L T4 DNA ligase, 1.5 U/ μ L Bsal). The reason we doubled the tile concentration in strategy 2 is because the direct activation strategy is more dynamic and higher tile concentration can benefit the nanotube assembly. Although the tile concentration is doubled, the final yield of the nanotube is more than thrice of the yield from strategy 1. We had discussed this in our main manuscript, which is also show below:

“Concurrently, the average length of DNTs per square micrometer reaches to 0.13 μ m/ μ m² (Fig. 6g), which is thrice the yield by ATP-fueled DSD cascades above (ca. 0.04 μ m/ μ m²), although the concentration for the assembling tile is only doubled. Histograms show length distribution of the DNTs (Fig. 6c; Supplementary Fig. 8). Compared to the system controlled by DSD cascades (above), shorter DNTs are observed at the DySS, which is attributed to the more dynamic nature of the DNTs due to direct restriction on the DNTs and also a higher ratio of Bsal to cleavable sequences (1.5 U/ μ L to 1 μ M versus 1.5 U/ μ L to 10 μ M). The higher DNT yield (as judged by the length per μ m²) is attributed to more efficient tile activation.”

- Can they comment about how the nanotubes are getting disassembled? From the end or it can start from any position in the middle?

Response: The nanotube can be degraded at any position in the middle when there is inhibitor strand deactivating the structures. We have further clarified this in our revised manuscript, which is also shown below:

Page 10: "It is worth noting that the DNTs can be degraded both from the end or any position in the middle when invaded by the inhibitor strands."

- The authors have mentioned the DNA nanotubes were imaged by diluting the solutions. What is the reason behind this?

Response: Dilution is needed for nice visualization (preventing overcrowding) in CLSM.

- The ATP fueled DNA cascade shown at 37 °C has a very low efficiency.

Response: This is not correct. We used the same enzyme concentration for the experiments at 25 and 37 °C. The reason for a lower increase of FI at 37 °C is due to higher enzyme activity for BsaI at 37 °C, which leads to more efficient cleavage of the ligated substrates, and, thus, more backward DSDs and lower increase of FI at its dynamic steady state. In principle, we should be able to achieve higher DSD at 37 °C by decreasing the BsaI concentration (reported for more simple ligation/restriction networks before). We have the discussion in our revised manuscript, which is also shown below:

"The shorter lifetime and lower FI increase compared to the ATP-fueled transient DSD cascades at 25 °C are attributed to higher BsaI activity at 37 °C.⁴⁰ To get a comparable ratio of DSD, one would need to increase T4 DNA ligation concentration or decrease the BsaI concentration."

Thank you for detailed comments to improve clarity.

Reviewers' Comments:

Reviewer #1:

Remarks to the Author:

The manuscript improved based on the comments of the reviewers. I have some specific responses below, but first is my general evaluation.

In their rebuttal and in the paper, the authors argue that the main contribution is the connection of multiple reaction networks/assemblies, each having been previously characterized by them or others. I agree, indeed this is the contribution. Our opinion differs on whether it is a major contribution.

Post revision, the substance of the paper and its suitability for a high impact journal hasn't changed, so my opinion hasn't either.

- I haven't learned much new from the design methods used to adapt the reactions modules.

The authors used ATP fueled reactions to direct strand displacement and colloid assemblies before [42]; here, they modified the strand displacement module to direct assembly of nanotubes. But a) the use of DSD to control nanotube behaviors have been shown in 2013 by Zhang and coauthors, and b) tile design and nanotube behaviors were also recently already demonstrated in recent work [35,36]. The modifications introduced are certainly sound but innovation is very limited. It boils down to, quoting the authors, "The ejected output strand [of the DSD reaction] is longer [relative to ref [42]]" so that specific component had to be adapted.

- I don't find major innovation in the experimental protocols.

- The behaviors achieved were also previously demonstrated in other papers.

To summarize, this is a solid execution of experiments that combine previously characterized parts and techniques achieving a predictable outcome. The paper does not offer major new insights in terms of design/engineering, protocols, or observed phenomena.

I would like to stress that I understand this project was laborious. But labor alone does not warrant publication in a high impact journal targeting a general audience.

I also found some issues with the responses and the corresponding revisions.

- "A relevant report is the autonomous control of DNA nanotube assembly via RNA transcription machinery by Franco and coworkers. This however uses sequential steps of reactions,36 and proceeds rather energetically downhill from the beginning. "

I think the authors are confused here or didn't read [36]. Beyond showing sequential assembly/disassembly, [36] shows the use of a molecular oscillator to direct nanotube assembly. I don't understand how this is "rather downhill". If the authors refer to exhaustion of fuel (NTPs, RNA), then this is true also for the system presented here, which eventually runs out of ATP.

- "The reviewer is not correct in stating that these strategies are identical. There are important conceptual differences on the design of the reaction network, the thermodynamics and in the experimental realization."

I regret writing a sentence that wasn't specific enough. I have read both papers and I do understand the upstream reaction networks are different.

My comment referred specifically to the strategy of using inhibitor/deactivator and activator that control assembly by binding to one of the tile sticky ends through a toehold, which is basically identical to the invasion/anti-invasion idea in [36]. Changes made in this paper are minor.

As to the difference in initial conditions (nanotubes pre-assembled or not), that is also a minor point. The assembly/disassembly in [36] was demonstrated in cycles, showing that it is possible to start with inhibited tiles and later activate them and vice versa.

- "The design of the nanotube in our figure certainly relates to ref.36, but it is not copy/paste."

While I agree it is not copy/paste, I will attach a snapshot from [36] for the editors to judge the similarity of the designs and the figures.

I have a final comment on the writing style: to me, most of the paper and the rebuttal reads like a "big word" salad.

In my personal opinion, an excess of jargon and complex words comes across as an attempt to overcompensate lack of substance.

Reviewer #2:

Remarks to the Author:

The authors addressed my comments and carefully revised the manuscript. I suggest the publication of this manuscript.

Reviewer #4:

Remarks to the Author:

The authors report growth of DNA nanostructures via coupled CRNs. While one example of CRNA-driven nanotube growth has been reported (Franco et al 2019), the approach in this paper is significantly different and more closely resembles how nature uses CRNs to drive the formation of higher-order structures. I have read the comments and the response to reviewer 3 and they are sufficient to allow publication of the work.

Reviewer #1 (Remarks to the Author):

The manuscript improved based on the comments of the reviewers. I have some specific responses below, but first is my general evaluation.

In their rebuttal and in the paper, the authors argue that the main contribution is the connection of multiple reaction networks/assemblies, each having been previously characterized by them or others. I agree, indeed this is the contribution. Our opinion differs on whether it is a major contribution. Post revision, the substance of the paper and its suitability for a high impact journal hasn't changed, so my opinion hasn't either.

- I haven't learned much new from the design methods used to adapt the reactions modules. The authors used ATP fueled reactions to direct strand displacement and colloid assemblies before [42]; here, they modified the strand displacement module to direct assembly of nanotubes. But a) the use of DSD to control nanotube behaviors have been shown in 2013 by Zhang and coauthors, and b) tile design and nanotube behaviors were also recently already demonstrated in recent work [35,36]. The modifications introduced are certainly sound but innovation is very limited. It boils down to, quoting the authors, "The ejected output strand [of the DSD reaction] is longer [relative to ref [42]]" so that specific component had to be adapted.

- I don't find major innovation in the experimental protocols.

- The behaviors achieved were also previously demonstrated in other papers.

To summarize, this is a solid execution of experiments that combine previously characterized parts and techniques achieving a predictable outcome. The paper does not offer major new insights in terms of design/engineering, protocols, or observed phenomena.

I would like to stress that I understand this project was laborious. But labor alone does not warrant publication in a high impact journal targeting a general audience.

I also found some issues with the responses and the corresponding revisions.

- "A relevant report is the autonomous control of DNA nanotube assembly via RNA transcription machinery by Franco and coworkers. This however uses sequential steps of reactions,³⁶ and proceeds rather energetically downhill from the beginning."

I think the authors are confused here or didn't read [36]. Beyond showing sequential assembly/disassembly, [36] shows the use of a molecular oscillator to direct nanotube assembly. I don't understand how this is "rather downhill". If the authors refer to exhaustion of fuel (NTPs, RNA), then this is true also for the system presented here, which eventually runs out of ATP.

- "The reviewer is not correct in stating that these strategies are identical. There are important conceptual differences on the design of the reaction network, the thermodynamics and in the experimental realization."

I regret writing a sentence that wasn't specific enough. I have read both papers and I do understand the upstream reaction networks are different.

My comment referred specifically to the strategy of using inhibitor/deactivator and activator that control assembly by binding to one of the tile sticky ends through a toehold, which is basically identical to the invasion/anti-invasion idea in [36]. Changes made in this paper are minor. As to the difference in initial conditions (nanotubes pre-assembled or not), that is also a minor point. The assembly/disassembly in [36] was demonstrated in cycles, showing that it is possible to start with inhibited tiles and later activate them and vice versa.

- "The design of the nanotube in our figure certainly relates to ref.36, but it is not copy/paste." While I agree it is not copy/paste, I will attach a snapshot from [36] for the editors to judge the similarity of the designs and the figures.

I have a final comment on the writing style: to me, most of the paper and the rebuttal reads like a "big word" salad.

In my personal opinion, an excess of jargon and complex words comes across as an attempt to overcompensate lack of substance.

Response: We thank the reviewer for his/her time in reviewing our paper. At this point we respectfully disagree with the reviewer.

The improvement of our initial ATP-fueled DNA strand displacement to eject longer strands is not trivial, but rather a significant innovation that broadens the application of such networks for DNA nanotechnology. Although DNA strand displacement-regulated DNA nanotube assembly was known, we rather chose it as a model system to show proof-of-concept for ATP-fueled hierarchically concatenated self-assembling systems. We provide *indirect* and *direct* strategies to control the assemblies. These are versatile concepts shown in significant breadth and depth.

Critically, we emphasize concatenation and communication of distinct chemical reaction networks for controlling dynamics of hierarchical structures. We believe this is a step forward also in synthetic biology for making artificial fuel-driven hierarchical self-assembling systems, as the present strategy more closely resembles how nature use chemical reaction networks to control higher-order structures. The differences to the Franco paper are well appreciated by reviewer #3/4.

As for the differences to Franco's paper, it is important to focus on the signal (B) and the process. Franco's paper uses a RNA signal to predominantly degrade formed nanotubes. The RNA signal is formed by T7RNAP from NTPs, the RNA is degraded by RNase H, allowing nanotube reassembly. The process for the signal is $A(\text{NTPs}) \rightarrow B(\text{RNA Signal}) \rightarrow C(\text{Waste})$. The signal is lost. The reviewer is right that oscillators are coupled upstream, and slight intermediate assembly/disassembly peaks appear.

We use ATP as a fuel to generate signal B. Signaling B molecule is energetically uphill and recovered again. This follows an $A \rightarrow B \rightarrow A$ process. The signal can be reactivated/reused by a fuel that is of no relevance to the nanotube formation (unlike RNA transcription based on NTP fuel). Additionally, we show rather transient generation of nanotubes. These network and system differences are conceptually significant, and also well appreciated by the other reviewers.

We clarified this with a small addition in the introduction.

In terms of the DNA nanotube design, we properly cited the paper and asked for the rights for reusing and adaptation of the figure.

Reviewer #2 (Remarks to the Author):

The authors addressed my comments and carefully revised the manuscript. I suggest the publication of this manuscript.

Response: We thank the reviewer for his/her time.

Reviewer #4 (Remarks to the Author):

The authors report growth of DNA nanostructures via coupled CRNs. While one example of CRNA-driven nanotube growth has been reported (Franco et al 2019), the approach in this paper is significantly different and more closely resembles how nature uses CRNs to drive the formation of higher-order structures. I have read the comments and the response to reviewer 3 and they are sufficient to allow publication of the work.

Response: We thank the reviewer for his/her time.